# WinSyn: A High Resolution Testbed for Synthetic Data

## Abstract

We present *WinSyn*, a dataset consisting of high-resolution photographs and renderings of 3D models as a testbed for synthetic-to-real research. The dataset consists of 75,739 high-resolution photographs of building windows, including traditional and modern designs, captured globally. These include 89,318 cropped subimages of windows, of which 9,002 are semantically labeled. Further, we present our domain-matched photorealistic procedural model which enables experimentation over a variety of parameter distributions and engineering approaches. Our procedural model provides a second corresponding dataset of 21,290 synthetic images. This jointly developed dataset is designed to facilitate research in the field of synthetic-to-real learning and synthetic data generation. WinSyn allows experimentation into the factors which make it challenging for synthetic data to compete with real-world data. We perform ablations using our synthetic model to identify the salient rendering, materials, and geometric factors pertinent to accuracy within the labeling task. We chose windows as a benchmark because they exhibit a large variability of geometry and materials in their design, making them ideal to study synthetic data generation in a constrained setting. We argue that the dataset is a crucial step to enable future research in synthetic data generation for deep learning.

## 1 Introduction

We describe a dataset for the purpose of advancing research in synthetic data generation for synthetic to real transfer and generative modeling. We overcome several limitations in previous data collection efforts for training machine learning systems to do 3D building modeling and reverse engineering of architecture. First, windows exhibit a high degree of variability in design and appearance characteristics, and yet also tend to have many symmetries that must be captured, making them more challenging than commonly used datasets for driving or faces. Furthermore, they have many thin

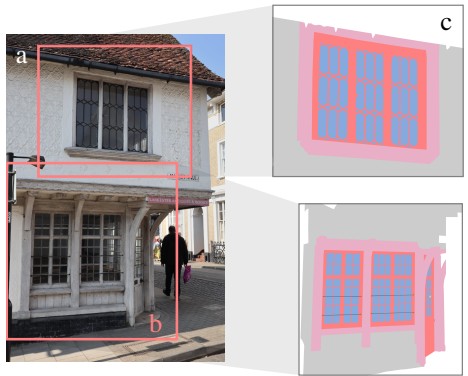 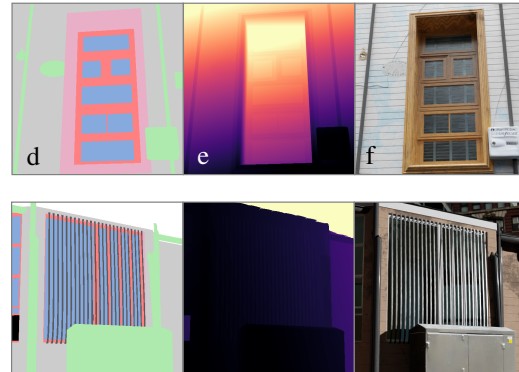

Figure 1: Photographers in 28 geographic regions capture real-world photos (a) of windows that are cropped (b) to single windows, which are then labeled (c). Synthetic windows are rendered giving color (f) and labels(d), while other passes such as depth (e; normalized per-image) are also possible.

structures (Fig. 1b) and reflections. The general trend is that solutions for generative modeling (diffusion or GANs) benefit from large datasets and that discriminative models such as segmentation methods may need to train for many iterations to capture fine details, but risk overtraining if the dataset is small. Our proposed dataset includes 75,739 images, which is similar to the sizes of the well-known CelebA-HQ dataset and the FFHQ dataset. This amount of data is sufficient to train high-quality generative models. Second, we would like to have high-quality and high-resolution images. Many modern images are captured and stored at resolutions of 4K or higher. A large portion of the images in our dataset have between 4K to 6K resolution which we ensured by restricting the cameras that were eligible for taking the images. Third, some datasets are captured by scraping the web but do not maintain copyright or ownership of the images. This can cause issues as data distributed over links disappears over time. Longterm availability of the dataset is more reliable if we own the rights to all the images. We, therefore, organized image collection from many subcontractors from around the globe. Fourth, we would like to select a domain that is especially suited for research in synthetic data generation. Synthetic data generation is a fascinating topic, but it generally has a large room for improvement. Architectural data, such as windows, is ideally situated for this, as many procedural models exist and are used in mapping, social and energy research applications, cinema, and video games. We selected the domain of windows for multiple reasons. The domain has sufficient variability, but it also has many constraints. We relay additional details of this discussion in the next paragraph. Fifth, we need a benchmark task that can be used to evaluate the success of synthetic data generation and training on synthetic data. For this benchmark task, we selected segmentation and we contracted a labeling company to provide segmentation labels for a subset of the window images. We also provide the corresponding dataset splits to evaluate different aspects of synthetic data. Finally, we wanted to have a set of images that can be used for feature extraction from raw images.

Windows have a large variability in structure. They appear in different countries, showcase different elements, materials, styles, and shapes. Nevertheless, we were confident that we could model the geometry and materials procedurally in high quality. The most complex aspects of windows are the glass elements, building interiors, and reflections. Windows are also important and common in urban images. We consider them to be a stepping stone to complete architectural models. By contrast, existing work (Wood et al.) had amazing success using artists to tackle the domain of humans. Unlike windows and other architectural elements, humans change shape but have comparably low variation in their overall structure and anatomy - whereas windows come in a variety of completely different structural configurations and textures. Architectural elements like windows have varying types and numbers of elements that interact in context-dependent ways – for example, it can be challenging even to determine if someone is looking at a single window vs multiple windows in some of the examples shown in Appendix Fig. 1, left. They also contain many similar, but confounding elements - such as reflections that mimic actual components, and design features like muntins that could be confused with divided lites. Additionally, there is semantic ambiguity in identifying specific components like sills, frames, and jambs, especially when they blend into the architectural context as seen in some instances in Fig. 2. On top of this, they exhibit a variety of regularities, such as symmetrical patterns, that are crucial to capture accurately; any deviations in spacing or symmetry are readily noticeable. These geometrical constraints, including thin structures and sharp corners, are particularly challenging for machine learning algorithms to model and represent. These complexities make windows a particularly challenging domain.

We also considered entire cityscapes (Dosovitskiy et al., 2017), but they are too complex to model realistically. Cityscapes would require realistic human models, car models, vegetation, and buildings as highly nontrivial modeling sub-problems. We hypothesized that it would be possible to create photo-realistic procedural models of windows in an academic setting, and it should be possible for others to compete on the tasks we set out. In spite of the narrower domain, we still observe a large gap in segmentation performance between real labeled images and synthetic labeled images. It is our belief that solutions that bridge the gap for windows may be extended to larger domains.

While our dataset was mainly motivated by the idea to improve synthetic data generation, we also support new research on a variety of different tasks. In summary, we make the following contributions: (1) We provide a 4K resolution dataset with 75,739 real-world images of windows (in some case multiple windows in an image) from around the world. Unlike previous datasets, these images are 4K×6K pixels cropped to 89,318 individual windows, providing an unprecedented level of detail for architectural image data. (2) We provide segmentation labels for 9,002 of these images.

All images and labels will be released. The URL will be released upon acceptance. (3) We propose a procedural model for windows and highlight design choices that are important for generating synthetic data for machine learning. (4) We introduce a dataset of 21,290 synthetic window images, including a wide variety of features that we have observed in real imagery. (5) We explore the importance of many features of a synthetic dataset in relation to the segmentation task using a wide variety of different experiments and ablations over our synthetic dataset.

## 2 RELATED WORK

Several datasets of architectural imagery have been created in the past, enabling applications such as architectural style classification (Xu et al., 2014; Chen et al., 2021; Barz & Denzler, 2021), building functional use classification (Kang et al., 2018; Zhao et al., 2021), architectural heritage classification (Llamas et al., 2017), landmark identification (Philbin et al., 2007), or urban scene matching (Hauagge & Snavely, 2012). Of particular note are datasets that support image synthesis or image segmentation. The FaSyn13 dataset (Dai et al., 2013) collected 200 facade images for the purpose of texture synthesis; but this is not enough data for modern generative models. The LSAA dataset (Zhu et al., 2022) contains 199,723 images of facades and 516,000 cropped images of windows. This is enough images, however, the resolution of each cropped window varies greatly, with the majority less than 100 pixels in the longest dimension. While the dataset we propose has fewer windows than LSAA, it is the highest resolution dataset of window images we are aware of at an average resolution of 4,000 pixels per side for cropped windows.

Several architectural image datasets exist that support semantic segmentation or facade parsing. The Graz dataset (Riemenschneider et al., 2012) contains 50 rectified images, and the eTRIMS datset (Korč & Förstner, 2009b) contains 60 non-rectified images. Without special "low shot" techniques, these datasets are not large enough to train modern deep-learning systems. The CMP-Facade dataset (Tyleček & Šára, 2013b) contains 606 images, with about half of them fairly high resolution (1024 pixels on the long edge) but limited diversity of image locations. The LabelME-Facade dataset (Brust et al., 2015a) has the largest number of images at 945, with each image varying in size between 512 and 768 pixels on a side. However, these datasets do not have fine-grained labels for windows, and neither the number of images nor the resolution is ideal for training the latest image segmentation methods. With 9,002 labeled images at four times the resolution of these datasets, our proposed dataset of real-world images is an order of magnitude larger.

Several authors have attempted to use synthetically generated data to bootstrap performance on real images. This approach seems to work best in domains where the human annotation is not directly feasible, such as reinforcement learning, especially for driving applications (Dosovitskiy et al., 2017), depth or optical flow (Butler et al., 2012; Gaidon et al., 2016), or 6DoF pose estimation for robotic grasping or manipulation (Tyree et al., 2022; Hodaň et al., 2020; Kaskman et al., 2019).

Infinigen (Raistrick et al., 2023) is a good example of a procedural model, however there is no validation of the effectiveness of the model for machine learning tasks. Of particular note is the SynthIA dataset (Ros et al., 2016), a driving dataset built on video game technology specifically designed to support semantic segmentation in urban environments. A very large engineering effort went into this, as well as CARLA, and we believe that reproducing such high-quality synthetic data is out of reach for most academic teams. Similar to our dataset, SynthIA aims at pushing the envelope to use synthetic data to improve computer vision even for problems where large human-annotated datasets (KITTI (Menze & Geiger, 2015), LabelME-Facade (Brust et al., 2015b), Camvid (Brostow et al., 2008)) already exist. Similar to our findings, they are able to get some results from purely synthetic data but they cannot out-compete even relatively small real-world labeled images, but by combining synthetic and real at a 4:6 ratio they obtain their best results. However, unlike SynthIA, our segmentation challenge is more constrained (only windows) and we think would require fewer resources for academic researchers to develop competing procedural models for synthetic training. Simultaneously domain transfer is more challenging due to the amount of variety and complex dependencies between architectural elements, whereas the categories of objects used in driving scenes are much clearer. In addition, the high resolution of the images we use makes the synthesis of realistic textures and precise object boundaries critical, and the higher-capacity segmentation models of today vs 2016 (when SynthIA was published) are more precise but may also be more likely to overfit synthetic data. Our dataset of real and synthetic imagery is unique as a high-resolution, voluminous

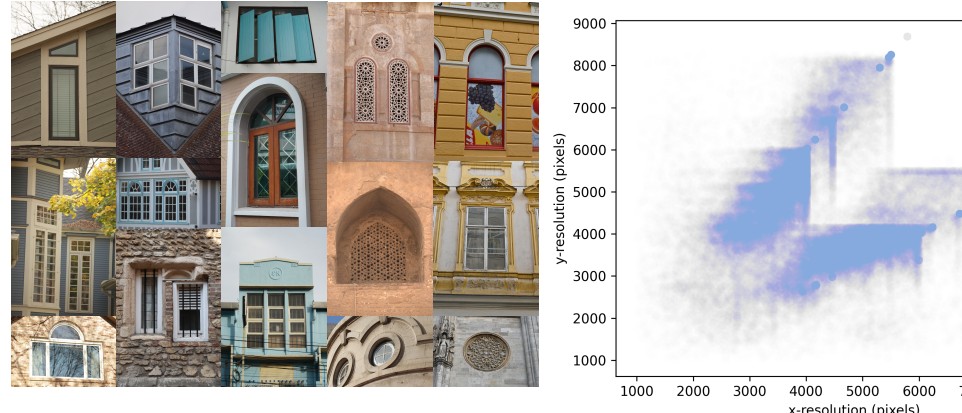

Figure 2: Left: Samples from the 75,739 photographs in the dataset. Each column shows a variety of examples of windows from different geographic locations. From left to right: Chicago (USA), Cambridge (UK), Bangkok (Thailand), Cairo (Egypt), and Vienna (Austria). The dataset has a variety of window shapes and architectural styles. Right: A scatter plot of the resolution of the 89,318 crops taken from these photos.

dataset and serves as a proving ground for synthetic to real training, image generation, and semantic segmentation tasks.

To our knowledge, the most successful work on using synthetic data for segmentations seems to be the 'Fake It Till You Make It' paper by Wood et al., which seems to report improvements in segmentation when a U-Net is trained using their synthetic data vs real image. However, in order to get these improved results, they used label adaptation. This is counter to our goals for using synthetic data, but it does highlight that the gap may be related to the synthetic vs human understanding of the label semantics. They do not report segmentation results without label adaptation, but they do show in their ablation study on landmark localization that label adaptation is critical for benefiting from synthetic data.

## 3 REAL-WINDOWS DATASET

We have collected the largest (over 75,739 photos), highest-resolution (up to $4K \times 6K$) dataset of images of windows that we are aware of, and unlike datasets formed by scraping the web or Flickr (Karras et al., 2019; 2017), we have ensured that we have the copyright ownership of each individual image[1]. When collecting photos of windows, it was important for us to have a diverse collection that included a variety of locations, viewpoints, and architectural styles. To achieve this, we decided to hire workers from several countries to ensure that the collection was representative of different regions and styles. While some of our own images are represented in the dataset, the overwhelming majority of the images were collected using freelancers, most of them hired on Upwork. Once hired, they collected the photos and submitted them for feedback to ensure that they met the following required standards. The photos must be framed in a way that showcases the window design and architecture without any unnecessary distractions. The framing should encompass the entire window and any interactions with the wall. As much as possible, foreground elements (vegetation, cars, wires) should be avoided. The photos must also be in focus, with sharp details and no blurriness or distortion of the window. The exposure of the images should be well-balanced, with reasonable image noise; therefore, most of the images were taken during the day. Workers should take care to avoid including people in private situations, personal items, and avoid personally identifying information in sensitive areas such as schools and hospitals to ensure that the privacy of individuals is respected. The images should show the window in a minimum of 4k resolution,

---

[1]In the papers we cited, the authors took great care to use images with permissive licenses and to maintain the correct attribution of each individual image. As we own the copyright to each image in our dataset this is less of a concern.

Figure 3: Examples of the labels used to annotate our data. Each instance receives its own polygon. The reader may wish to zoom into the figure for details.

although this was not strictly enforced, with a goal of 2k pixels across each window. Finally, The images should be taken with a professional-level SLR or Mirrorless camera. Images collected this way ended up costing between US $0.20-$0.50 each, depending on the contract. This is in addition to our own cost of quality control and managing the subcontracts. We hired 24 photographers in total over 12 months.

The diversity of image locations is indicated in Table 1 of the appendix, along with the number of images that include semantic segmentation labels and RAW camera data. RAW camera data refers to the unprocessed and uncompressed image data captured by a digital camera's sensor. It contains all the information captured by the camera's sensor without any loss of detail or quality. This data may include higher precision (typically 12-14 bits rather than 8 bits per channel; see Appendix Figure 4) of radiometric color information that has not already been subject to de-mosaicing from a Bayer filter, color space conversion, or any other in-camera image signal processing (ISP) followed by JPEG compression. We hypothesize that this information may allow for improved performance on certain tasks, however, each camera may use its own proprietary RAW format, and it is much more voluminous and harder to distribute, load, and manipulate than compressed imagery. Our dataset includes 6,666 photographs with both the RAW camera information and segmentation masks.

Many images included multiple windows. We manually annotated each image to a region that includes a single window in the center, along with any portion of the wall that may have been adapted to the window (such as brickwork or molding) and a small portion of the wall on all four sides of the window. Due to cropping, window images are in a variety of sizes, as shown in Fig. 2. We store the original images and the cropping information separately and generate a cropped version of the dataset on demand.

## 4 SYNTHETIC DATA

We designed a pipeline combining procedural modeling, texturing, and rendering to create 21,290 synthetic images of windows and corresponding labels, as detailed in Appendix Fig. 2. Our system is Python-based and uses Blender (Community, 2023). Our pipeline sequentially generates geometry, applies textures, sets lighting, and configures cameras. Rather than render a realistic subset of real-world data we aim for higher model diversity, allowing implausible results and akin to domain randomization (Tobin et al., 2017). In order to handle reflections and shadows, windows are rendered in context within buildings, streets, and urban elements; details are in Appendix Fig 8. Further details are in section 5 of the appendix.

The procedural model relies on two types of Split Grammar (Wonka et al., 2003). We employ the CGA language (Mueller et al., 2006) for facades, windows, and urban structures. Window shapes use Bézier splines for various geometries (e.g. arched or circular windows). A second grammar splits the windows into individual panes with distinct extruded profiles. Pseudo-random distributions guide the generation process, and parameters stored in text files for reproducibility. Each parameter is sampled from its own distribution, and the number of parameters depend on the random sequence of steps taken by the procedural model. In practice, the pipeline is parameterized by 216 to 21,735 variables, enabling a wide range of scene variations.

Textures include static images, but are primarily procedural shaders (Burley & Studios, 2012), controlling materials like wood, brick, or glass. To add realism, we captured exterior clutter like signs and trash cans through LiDAR and RGB scanning (Appendix Fig. 9). Lighting is a blend of skybox

| Label | Images Using | Area % |
|---|---|---|
| wall | 8907 | 43.02% |
| window pane | 8362 | 22.58% |
| wall frame | 8697 | 14.91% |
| window frame | 8681 | 9.71% |
| unlabeled | 2994 | 3.09% |
| shutter | 931 | 2.56% |
| balcony | 973 | 1.08% |
| misc object | 2357 | 1.07% |
| blind | 375 | 0.75% |
| bars | 679 | 0.68% |
| open-window | 977 | 0.55% |

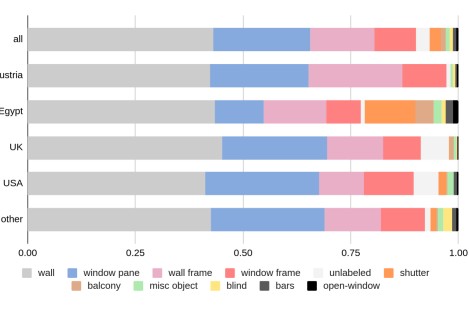

Figure 4: Left: The labels used, their frequency of use, and percentage by area for the square-cropped dataset used for our experiments. We note that the dataset is a mix of well-used labels such as wall and less-used ones, such as door or bars. The 'unlabeled' category contained areas beyond the building (e.g., sky, streets) and a much smaller number of ambiguous areas where we could not reach a decision on how to label a feature. Right: the label fraction for the different geographic partitions of the labeled data

emission, direct sun-lamp, and optional interior sources. The camera is strategically positioned in order to capture the entire from a predominantly frontal view.

## 5 SEMANTIC SEGMENTATION

We propose semantic segmentation as a suitable benchmark task for evaluating the quality of synthetic data. The idea is to train a segmentation method using synthetically generated data, train the same method using manually labeled real-world data, and then use the difference in performance as a way to evaluate the quality of the synthetic data.

Existing segmentation datasets often have a single label for 'window', possibly with labels for additional adornments such as 'shutter', 'blind', or 'sill' (Tyleček & Šára, 2013a; Korč & Förstner, 2009a; Brust et al., 2015b). We found that these labels are difficult to apply consistently in light of the variety of architectural styles encountered in the imagery. Some elements in the images had dual or ambiguous labels, and sometimes an identical element had a different semantic role in different images or relative to different objects (e.g., a higher window's 'sill' is a 'lintel' of a window below). Among other things, this made it challenging to describe a consistent set of categories that could be communicated to the multiple individuals who label the data. Instead, we developed a set of twelve semantic categories that could be applied to most images, including labels such as 'window-pane' (transparent openings), 'window-frame', 'wall frame' (the portion of the wall adapted to a window), and others. Each instance was annotated by its own polygon. Often it was clear that an object, such as a wall or a window pane, was divided into separate parts based on changes in material, texture, or depth. In these cases, the individual parts were given separate polygons with the same label, however, this was not consistently applied, in part due to different interpretations of the images by annotators. In many cases, some element of the image would be ambiguous, and no single label could be determined with certainty. Examples include a painting of a window or objects in the background that are out of focus. These objects were explicitly marked as 'unlabeled'. Examples of images annotated using these labels are shown in Fig. 3, and label frequencies are shown in Table 4. The complete set of labels is described in the supplemental. Each image was manually annotated, and then carefully reviewed for quality and consistency. The resulting set of 9,002 annotated images was acquired at a cost of about US $3.90 per image – this cost is manageable to some degree, but the price of annotation is significantly higher than the cost of image acquisition, so any method that can reduce the number of labels is important, especially for segmentation which is labor-intensive.

|  |  | test | | | | | | |
|---|---|---|---|---|---|---|---|---|
|  |  | global | Austria | Egypt | UK | USA | other | synthetic |
| train | global | 53.79 | 58.20 | 56.85 | 42.64 | 50.16 | 54.74 | 31.21 |
|  | Austria | 39.49 | 58.23 | 28.36 | 34.40 | 35.67 | 41.32 | 21.87 |
|  | Egypt | 47.51 | 38.32 | 61.85 | 33.02 | 38.01 | 49.75 | 31.34 |
|  | UK | 37.76 | 40.68 | 25.23 | 38.56 | 35.39 | 37.64 | 28.51 |
|  | USA | 41.08 | 42.50 | 27.39 | 33.30 | 50.08 | 39.39 | 27.02 |
|  | other | 51.22 | 48.54 | 39.47 | 37.44 | 41.33 | 52.74 | 28.39 |
|  | synthetic | 31.23 | 29.53 | 29.33 | 32.15 | 34.17 | 35.02 | 62.12 |

Table 1: mIoU for different splits of the real labeled data on the segmentation task. Trained on $n = 1024$, tested on $n = 300$. *global* is a mixture of all the real data; *other* data is from locales outside of Austria, Egypt, UK, or USA. Here our synthetic model is similarly trained on $n = 1024$ samples from our baseline synthetic model.

# 6 ANALYSIS

## 6.1 GENERALIZATION RESULTS WITH BEiT

We use BEiT v2 (Peng et al., 2022; Bao et al., 2022) as a baseline model for image segmentation because it is near the state of the art for segmentation at this time, and a reliable implementation of it exists which can be trained quickly in order to conduct experiments. BEiT adapts the concept of masked language modeling from BERT (Devlin et al., 2018) and applies it to images by self-supervised masked image pretraining, wherein portions of an image are masked-out and BEiT predicts the missing piece. BEiT is then refined for specific tasks such as semantic segmentation. BEiT has a 'base' model with 86M parameters and a 'large' model with 300M parameters. In all of our experiments, we fine-tune a BEiT 'base' model that was pre-trained on ImageNet-1k, trained on ImageNet-21k, and fine-tuned on ours. We evaluated the mIoU over 10 labels, excluding "unlabelled" category. All images are 512 pixels square.

A key concern is the generalizability of models trained on architectural datasets, especially since it can be challenging to collect data from a diverse set of locations. To test this, we conducted the following experiment. We grouped our images into five groups; from Austria, Egypt, UK, USA, and "other" locales. For each group, we sampled 1,024 images for training, and a further 300 images for testing.

We are particularly interested in how well segmentation performance scales with the size of our training dataset. We established an experimental set of $n = 4.9k$ images (which we use for the remainder of the paper; except where noted), to allow training on $n = 4096$ images and testing on the remaining 804. In Table. 1, we found that synthetic data yields performance ranging from 29.33 to 35.02, whereas training on different cities produces a wider range, from 25.23 to 49.75. Although training on individual cities outperforms synthetic data when evaluated on the global set, these figures are influenced by the inclusion of each city in the global data. On the positive side, the performance of synthetic data can even be better than training on real data from a different country (England vs. Egypt). On the negative side, the difference on the global dataset is still much larger than desirable (53.79 vs. 31.21 mIoU).

Fig. 5 explores the impact of synthetic data on segmentation. From the tests, we draw the following conclusions: First, we see that training on 100% synthetic data already gives fair quality (mIoU 32.58). As more real data is added to the training dataset, the performance, improves until it reaches an mIoU of 44.45%, which is about 76% of the best mIoU obtained by using all the real-world imagery in the split (59.192) and it does so using only 3.7% of the real-world data. It is noteworthy that for larger amounts of real-world data, the synthetic imagery slightly harms performance. When all the real world and synthetic data from this split are used, the mIoU is 57.074, which is 96% as good as using only the real world imagery. This is to be expected if our synthetic data does not perfectly match the distribution of real images. Because synthetic data is low-cost and easy to create, we are able to generate large datasets quickly. Ideally, all training could be done on synthetic images and the costly real imagery could be used only for testing. However, our experiments show

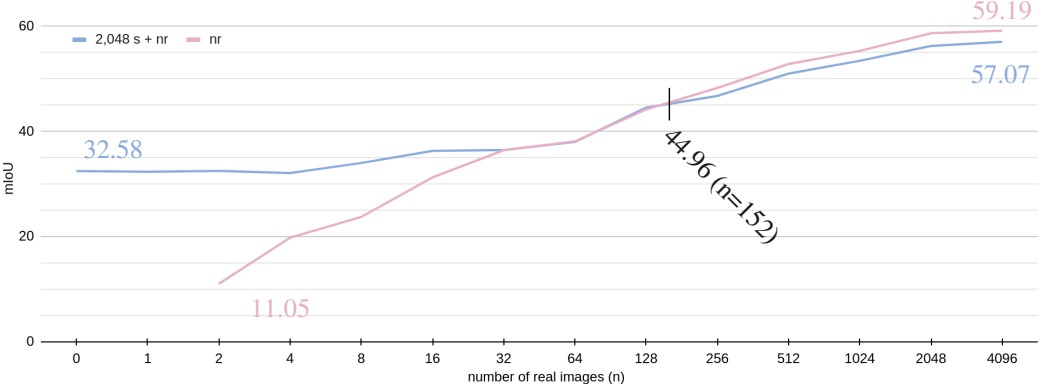

Figure 5: Varying real-world dataset sizes on mIoU with (blue) and without (red) an additional 2,048 synthetic samples. Intersection at $n = 152$ with mIoU of $44.96$ shows efficiency of synthetic data. At larger datasets, synthetic data reduces mIoU by approx 3.6% relative to real-world data.

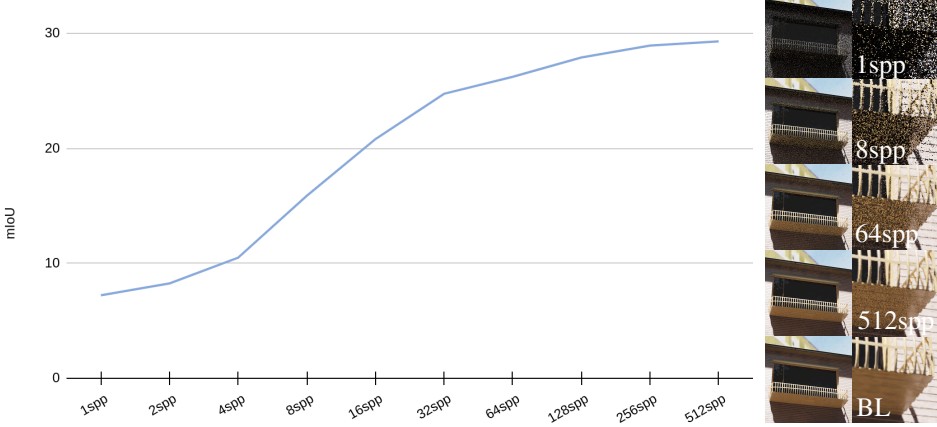

Figure 6: The impact of rendering samples per pixel (spp) on segmentation task accuracy. Left image column: example rendering. Right column: zoomed section of rendering. BL = baseline.

improvements are limited and that the number of synthetic images which are useful depends on the number of real images available.

## 6.2 PROCEDURAL MODEL VARIATIONS

We assess the impact of synthetic dataset variations on a segmentation model by comparing against our baseline model. Each variation has $2K$ training examples and uses the same geometry, lighting, and rendering settings as the baseline unless specified. To gauge variation importance, we either use mIoU's relative range as a percentage of the baseline or report Spearman's rank correlation($r_s$). Detailed results are in section 6 of the appendix and are summarized here.

**Rendering samples.** We evaluated the impact of samples per pixel (spp) on render quality, noting diminishing returns beyond 256spp (Fig. 6). Render times scale from 6.4s at 1spp to 85.2s at 512spp. A strong correlation ($r_s = 1, n = 10$) exists between spp and mIoU, with a 68% change in mIoU scores relative to baseline, underlining the importance of spp. In our experiments' no denoising was done; however, our baseline model used 256spp and a powerful neural denoiser.

**Materials.** We experimented with 9 material variations, including uniform gray, edge shaders, and random textures. Some tests used a single material for the entire scene, others assigned different

materials per object. Simplifying materials, such as rendering only the albedo channel, resulted in a significant performance drop. Fig. 12 shows that any material restriction led to at least an 18% mIoU decrease from the baseline.

**Lighting.** We examined 8 lighting model variations and their mIoU impact, as detailed in Fig. 13. Lighting models like albedo-only were crucial, while lighting-conditions such as nighttime had moderate importance. Conditions varied mIoU by 15.35% from baseline. Daylight-only training decreased baseline mIoU by 1.04% relative to baseline.

**Camera.** We experimented with the distribution of camera positions. These 6 variations used a simple model which sampled a camera position over a circle, of radius $r = \{0..48\}$ meters, truncated at the floor plane. The circle is positioned at 5 meters from the wall, directly in front of the window. All cameras have their field of view adjusted to the apparent window size. We observed very little impact on mIoU as $r$ changes; correlation was poor ($r_s = 0.14, n = 6$).

**Window Geometry.** We ran 7 tests varying window dimensions and shapes, including square and non-rectangular windows. The mIoU impact was minor, fluctuating by up to 5.7% relative to the baseline (1.86 absolute), with the best variation 1.2% worse relative to the baseline ($-0.04$ absolute). Small features, though noticeable to humans, had a weak impact on model performance.

**Labels Modeled.** In developing our procedural synthetic data generator, we prioritized labels by size, starting with *wall* and ending with *open-window* (Figure 5). This enabled assessment of mIoU at nine developmental stages. Adding smaller classes later showed diminishing returns and occasionally reduced label accuracy.

**Histogram Matching:** We investigated histogram matching (Gonzales & Wintz, 1987) to align the training data's distribution with that of the real-world training split. Unlike the unsupervised per-image histogram equalization (see Appendix section 6.2), this required supervision. The method led to a minor mIoU improvement, from 32.58 to 32.96 (1.17% increase).

**Label Adaptation:** Our focus is on reducing the need for labeled images. Despite this aim, we tested label adaptation as per Wood et al.. Though it doesn't minimize labeled image use, it significantly boosted mIoU from 32.58 to 41.55. This technique is akin to adding 64-128 real samples to our synthetic training set. See examples in Fig. 17 of the appendix. We believe that the technique of label adaptation may create an overly optimistic picture of synthetic data. It often indirectly uses real data as shape prior and it seems to do a lot more than simply adapting differences in labeling.

In summary, we can conclude that synthetic data can be very useful, but the gap between synthetic and real data is still larger than desirable, even in a constrained setting such as windows (53.79 vs. 31.21 mIoU on the global dataset). While we did spend an extensive effort to create a very high-quality procedural model, we believe that much additional work will be needed to understand and create synthetic data. We believe that our dataset can be the stepping stone to future progress.

## 7 CONCLUSION

We have introduced a new dataset of 75,739 photos (2.09 terapixels), 9,002 semantically labeled images (including RAW images), for applications like superresolution, 3D reconstruction, and generative modeling. We also presented a high-quality procedural model that closely approximates real-world variation, making it effective for image segmentation tasks. We systematically explored the effect of variations on our model on mIoU. The mIoU performance gap between our synthetic and real-world data is comparable to inter-city differences with largely different architectural styles. However, the difference between synthetic data and real data is still much larger than desired. This gap is not really explained by either our work or any other competing work. We, therefore, believe that research in synthetic data generation is important and that our dataset can be a significant contributor to future work in this area and to getting a better understanding of the critical differences between real and synthetic data. Our work contributes a sizable, versatile dataset that can be the basis of exciting and much-needed progress in the area of synthetic data generation for machine learning. The key value of our dataset is that as a new benchmark of simulated and real images it enables others to study the problem at the right level of complexity and that it will therefore be instrumental in making progress.

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
