# 1 PHOTO DIVERSITY

| Location | Images | Raw | Labeled | Crops | Labeled Raw |
|---|---|---|---|---|---|
| Algeria | 1714 | 1714 | 0 | 1795 | 0 |
| Argentina | 4153 | 4153 | 243 | 4610 | 243 |
| Armenia | 2135 | 2135 | 0 | 2143 | 0 |
| Austria | 9494 | 9494 | 2139 | 13966 | 2139 |
| Bangladesh | 1996 | 1996 | 64 | 2224 | 64 |
| Brazil | 2102 | 2102 | 35 | 2750 | 35 |
| Canada | 1999 | 1999 | 0 | 2115 | 0 |
| China | 2115 | 2114 | 0 | 2466 | 0 |
| Columbia | 663 | 663 | 0 | 685 | 0 |
| Cyprus | 2077 | 2077 | 93 | 2070 | 93 |
| Czechia | 2183 | 2183 | 102 | 2159 | 102 |
| Denmark | 233 | 0 | 78 | 254 | 0 |
| Egypt | 3873 | 3855 | 1500 | 4436 | 1496 |
| Germany | 4136 | 4136 | 861 | 6083 | 861 |
| Greece | 2045 | 2045 | 62 | 2085 | 62 |
| India | 1943 | 1934 | 0 | 2143 | 0 |
| Ireland | 2057 | 2057 | 58 | 2049 | 58 |
| Macedonia | 2024 | 2024 | 79 | 2050 | 79 |
| misc | 184 | 0 | 32 | 171 | 0 |
| Morocco | 2006 | 2006 | 0 | 2013 | 0 |
| Philippines | 1829 | 1828 | 0 | 2170 | 0 |
| Poland | 4072 | 4072 | 20 | 5851 | 20 |
| Saudi Arabia | 213 | 213 | 0 | 238 | 0 |
| Tanzania | 1597 | 1597 | 0 | 1644 | 0 |
| Thailand | 1009 | 1009 | 44 | 1052 | 44 |
| Turkey | 3027 | 3025 | 0 | 3116 | 0 |
| UK | 5541 | 481 | 2015 | 6760 | 0 |
| USA | 9319 | 8801 | 1577 | 10220 | 1370 |
| Totals | 75,739 | 69713 | 9,002 | 89,318 | 6,666 |

Table 1: Breakdown of the number of images by location, along with the number of images with labels, the number in RAW format, and the number of crops (rectangle around individual windows). Note that more than one window can be cropped from an image, and only the high-quality windows which are in-focus and large enough to clearly resolve have been cropped.

# 2 DATASET EXAMPLES

We provide examples of the photographs, labels, synthetic images, and RAW images in Figures 1, 2, 4, and 3, respectively.

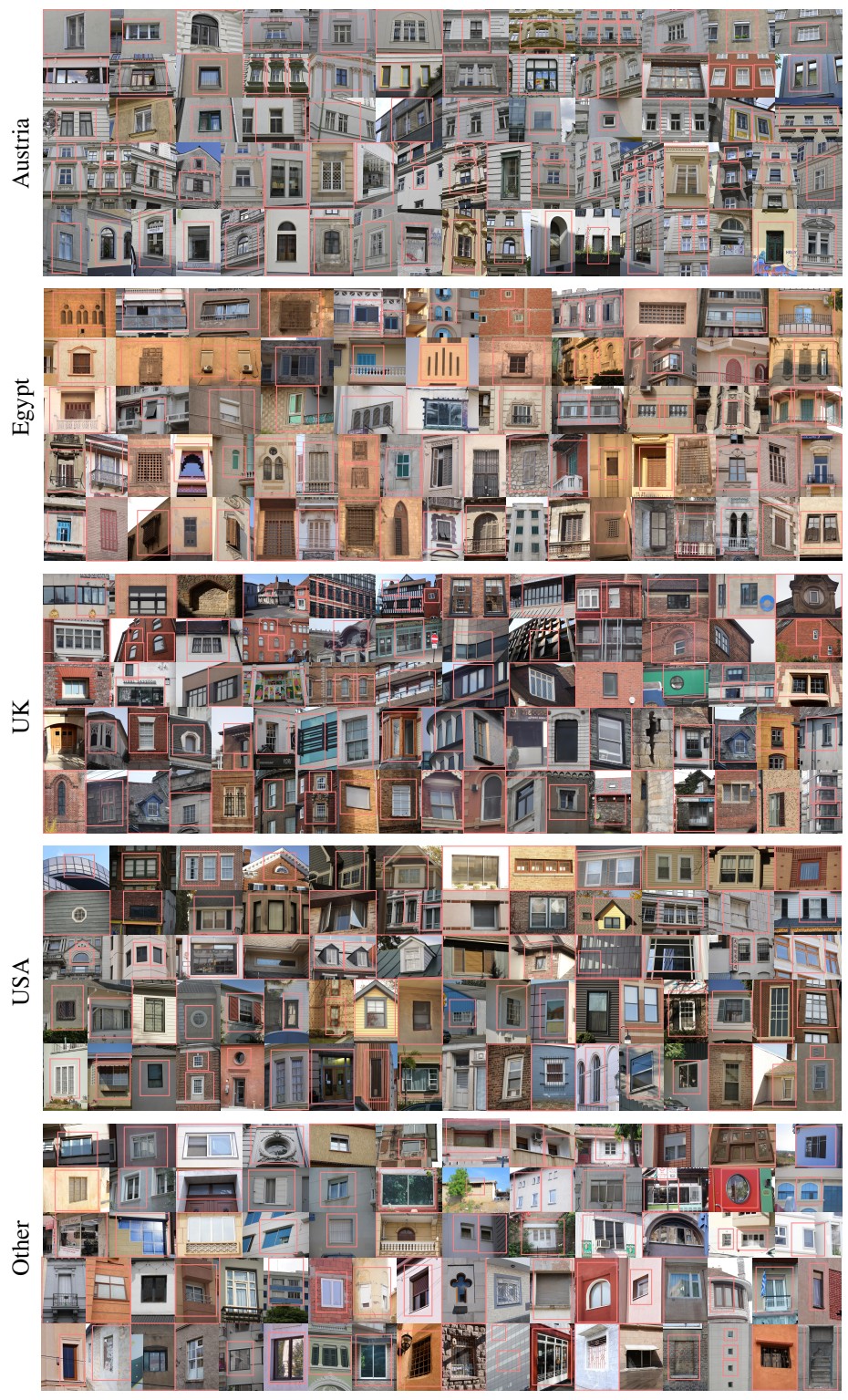

Figure 1: A random selection of our 75,739 window photos and their 89,318 crops (red boxes) from Austria, Egypt, UK, USA, and Other. Each partition contains at least 1,500 photographs. Zoom-in for easier viewing.

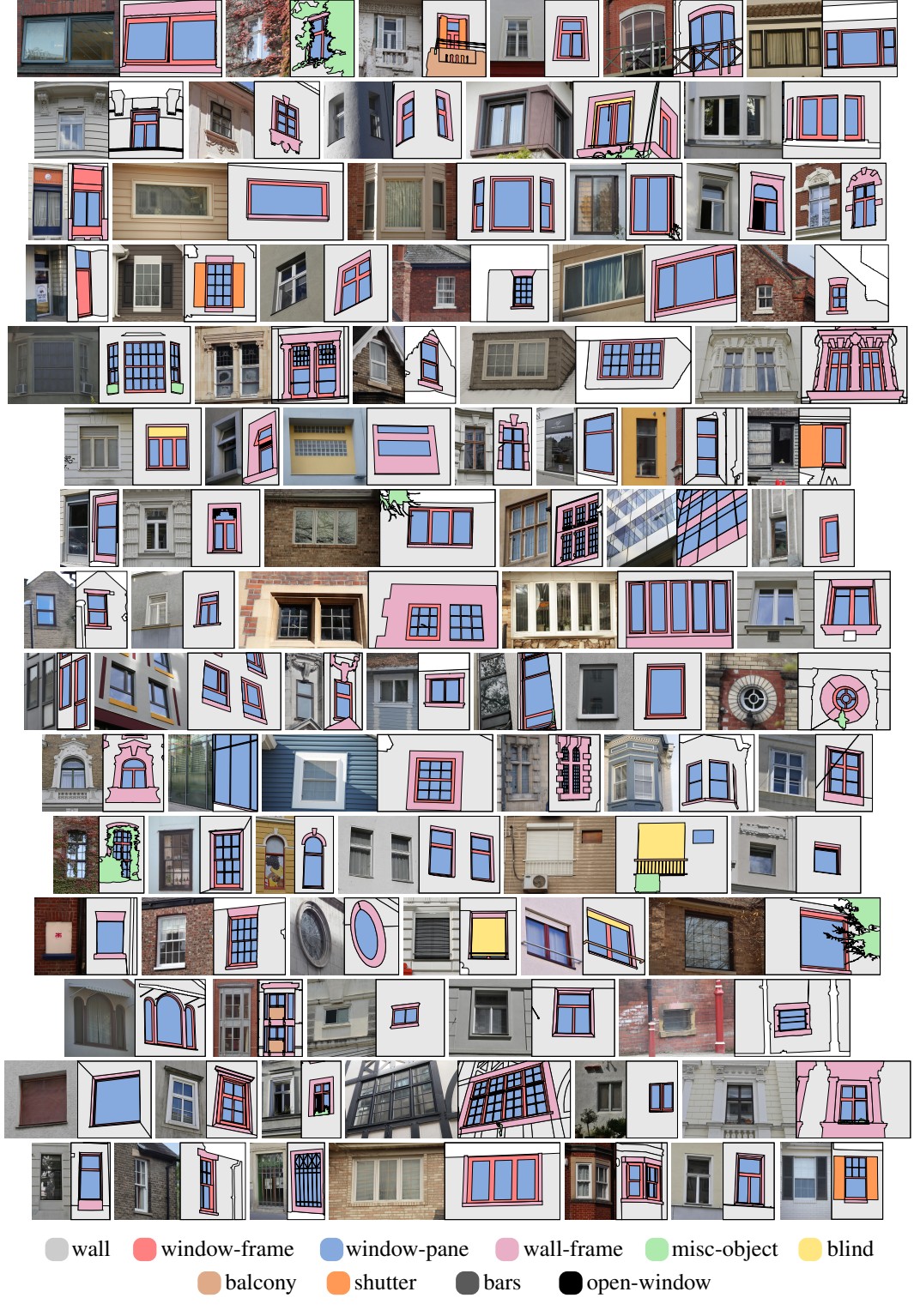

wall  window-frame  window-pane  wall-frame  misc-object  blind

balcony  shutter  bars  open-window

Figure 2: A random selection our 9,002 labeled cropped photos. Zoom-in for easier viewing.

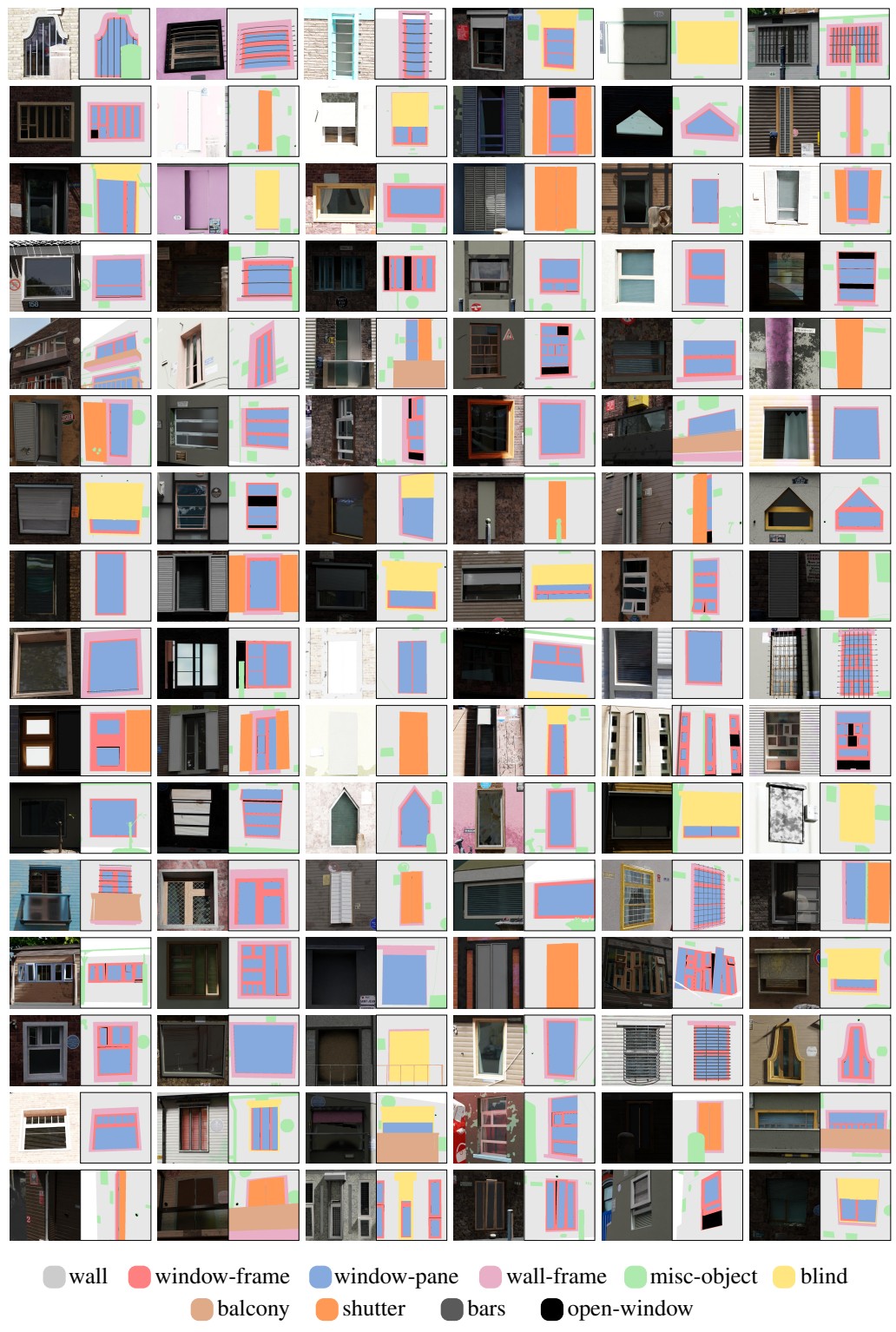

Figure 3: Random samples from the 21,290 in the synthetic window dataset showing color and label channels. Zoom-in for easier viewing.

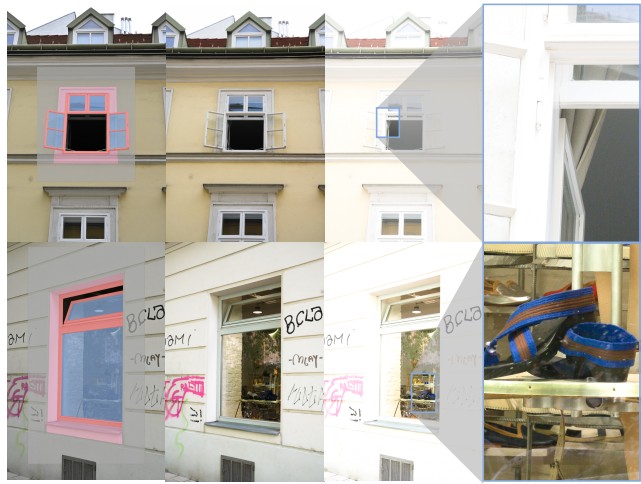

Figure 4: 6,666 of our photos contain the original RAW data as well as labels. Above: the default JPG image and labels (column 1), reprocessed RAW files for global brightness (2), reprocessed RAW files for a given region's (blue boxes) brightness (column 3), and a crop of the result showing the full resolution (4).

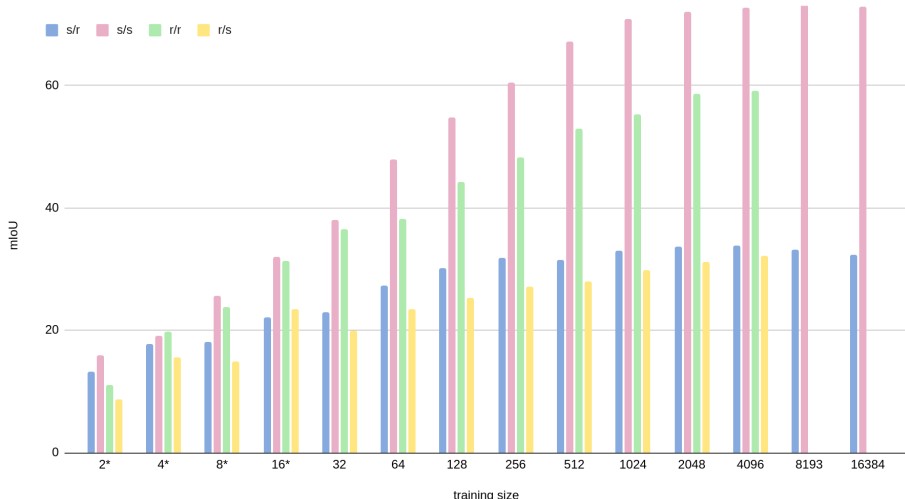

Figure 5: Performance with different amounts of synthetic and real data. s/r was trained on synthetic and tested on real. Models with a sample size below 16 were repeated 5 times with different samples from the test set. These are marked with an asterisk. All class mIoU excluding unlabeled. Test size was 4.9k.

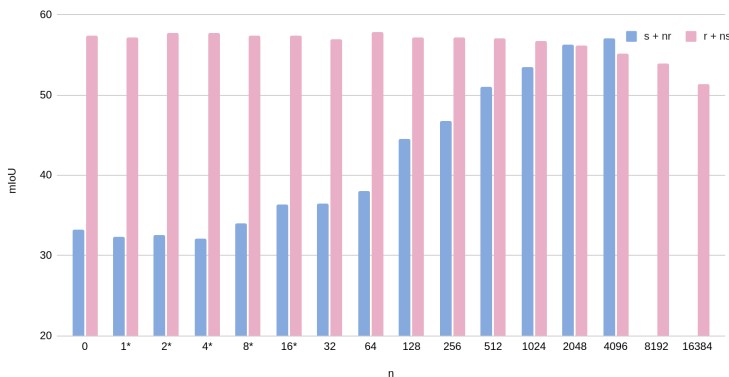

Figure 6: Segmentation mIoU with various mixes of real and synthetic data. The blue bars show the effect of adding $n$ real data to 2,048 synthetic images. The red bar shows the effect of adding $n$ synthetic data to 2,048 real images. Models with a sample size below 16 were repeated 5 times with a different selection of $n$. These are marked with an asterisk. All class mIoU excluding unlabeled. Test size was 4.9k.

## 3 DATASET EXPERIMENTS

We performed several experiments to determine the impact on segmentation task mIoU of the amount of data (Figure 5) and the mixture of real and synthetic data (Figure 6 ) on our dataset. Additionally, we explore the label quality at different mIoUs in Figure 7.

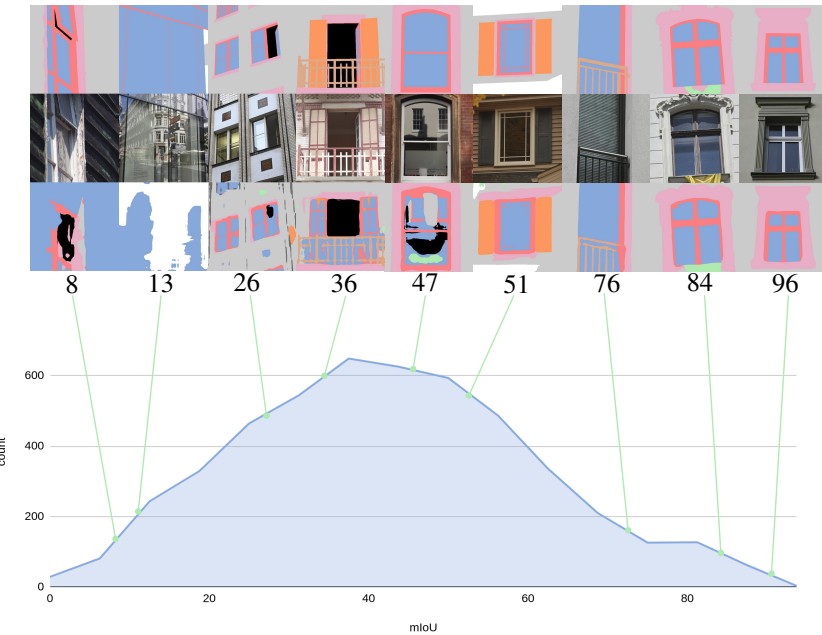

Figure 7: Examples of labeling results by mIoU. Top row: ground truth labels. Second row: photos. Third row: labeling results with mIoU. Below: Histogram of individual image mIoUs for the baseline model trained on $n = 2,048$ synthetic images. Note the mean of the per-image mIoU is higher that the whole dataset because samples may not contain all labels, in which case that label doesn't contribute to the mIoU.

## 4  LABELS

The following labels are used to annotate a subset of the cropped photos:

1. **window-pane**: Glass, painted "glass", opening (missing glass), mesh-screen, or repair (e.g., wood/brick covering a broken pane)

2. **window-frame**: Not part of the wall, part of the window, usually wood, metal, or plastic. Used infrequently for door frames.

3. **open-window**: The interior of the building.

4. **wall-frame**: Part of the wall which is adapted to the window. Each of the following should have this label, but be different instances.:

   (a) *window apron* (sill; part of the wall, below the window)

   (b) *window header* (lintel; part of the wall, above the window)

   (c) *wall frame* (part of the wall; decorates/supports the window)

   (d) *balcony base* (support; below balcony, holds up the balcony railings)

5. **wall**: Other parts of the wall

6. **shutter**: These are beside the window and swing sideways to protect and insulate the window.

7. **blind**: Exterior blinds. these are above the window and move down to protect and insulate the window.

8. **bars**: Fixtures protecting the window and frame.

9. **balcony**: Guard-rails, railings, balconette.

10. **misc-object**: In front of the glass: people, pot-plants, toys, junk, pipes, wires, trees, plants on the wall, alarm, unknown objects, misc. foreground.

Other guidance provided to labellers:

- Objects seen through or "inside the frame" of the main windows should be ignored. For example, if you can see another window through the main window then ignore it.

- Objects reflected (in the glass) should be ignored.

- Fine window structures (leaded glass, "fake" (plastic) leaded glass inside glass, security chain link fence, chicken wire) should be ignored. Larger structures (cast iron fence) should be labelled. If uncertain, ignore.

- Label other objects in front of the window as misc-object. Other objects (e.g., in front of the wall) may be left un-labelled or also labelled as misc-object.

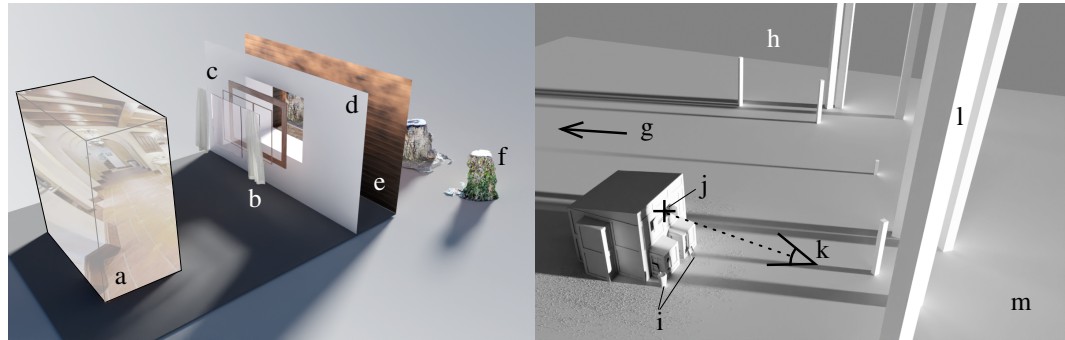

Figure 8: Left: A layered approach to window realism. The window is composed of nested layers of frame and glass (c) and is optionally dressed (b, curtains). Interior (d) and exterior (e) walls divide the inside from the outside. An interior-box (a) is the geometry onto which we project an interior panorama, while the exterior is decorated with street clutter (f). Right: Our environment rig. The building geometry lies on a floor plane (m) under a skybox (h). It is lit by the skybox and from a directional sun lamp (g), which may cast shadows from the urban canyon (l). The camera is positioned in front of the building (k), pointing towards the camera target (j) within the primary window.

## 5 PROCEDURAL MODEL DETAILS

This section provides additional details about the *baseline* procedural model used to create the scenes. Figure 8 provides an overview.

**Building mass.** The walls, roof, bay windows, and wings are created using the *Computer Generated Architecture* (CGA) (Mueller et al., 2006) procedural modeling language. The grammar we use creates a rectangle on the floor-plane between 3 and 9 meters wide and 3 and 5 meters deep. This is extruded up to create between 1 and 4 stories. Wings are optionally created from the side faces; bay-windows are extruded on the front of the mode. Hip, shed, or gable roofs are added to the top of these masses. The front of the building and bay walls are split to create walls and windows. Optionally, a timber-frame is created within these wall panels.

**Window** geometry is created within a rectangle proscribed by a standard CGA building grammar. Because CGA isn't able to model curves or extrude profiles, we use a second split grammar to create a variety of window shapes. The output of the grammar is a grouped hierarchy of Bézier splines, each assigned a profile. This allows the creation of complex windows for example where the window casing (around the edge of the window) has a different profile to the static panels, which have a different profile to the panels which move when the window opens. A single set of profiles is selected from 14 set available (Figure 16, right). Each spline is then assigned one profile within the set and filled with glass. The grammar sequences split rules to convert the input rectangle into groups within the hierarchy.

We have four classes of shapes (rectangular, trapezoid, circular, arched). Each of these classes has several sub-variants. For example, circular windows may become semi-circular or window arches may be straight or curve. Each shapes has a splits appropriate to that geometry (e.g., circular windows only split into quarters) that are applied with certain probabilities defined by the grammar. Some shapes split into each other; for example, the bottom of an arched window is a rectangle and shares the splits available.

To model open windows, parts of the this hierarchy may be translated (for sash or sliding windows) or rotated (for hinged windows) This is can be identified in Figure 3 by looking for the black *open-window* labels.

Window surrounds (sills, lintels, complete frames) are constructed from extruded profiles over splines. These decorate the wall and recessed area around the window. We use sections of the window-shape to describe the spline shapes; for example, a window lintel is the top section of a window-shape, which may be arched or circular. The spline is optionally extruded beyond the edge of the window.

There is a small probability (0.013) of creating *blind* windows (windows without a frame or glass) or a window without glass. These features were added after observing them in the dataset.

**Interiors.** Window-dressing is positioned inside the windows, and allows curtains, Venetian blinds, and wooden or fabric blinds. The soft dressings use a cloth simulation to create bunches and gather the material for different positions and window shapes. Behind the dressing we use a "interior-box" onto which we project an interior panorama randomly selected from a library. At night, this box emits light, simulating a lit interior room. Window dressing is only applied to the "primary window" - the one which the camera is pointing at.

**Lighting**. The scene has three light sources - a panorama providing omni-directional light based on the selected skydome background image, a sun adding a directionaly strong light-source, and an (optionally) lit interior material allowing interior lighting at night. The sun-lamp's light filters through an "urban canyon" that simulates the light passing around other buildings as shown in Fig 8. This is a stochastically generated area of cuboids with mixed size and height behind the camera. The geometry allows indirect light paths and shadows to fall onto the building, adding image features similar to those to in the ground-truth such as shadows from trees, telegraph poles, and other buildings. The urban canyon is not directly visible to the camera or in reflections.

The sun is rotated with an azimuth between -90 to 90 degrees, with increased chance of being near either extreme. This increases the chance the sun direction being parallel to the wall and creating large "glancing" shadows. The altitude of the sun is normally distributed with $\mu = 40, \sigma = 28$ degrees. The size of the sun the also varies - creating the effect of light though different sky conditions.

With a probability of 0.05, we create a night seen. The light emission from the exterior skydome and sun is reduced, and the interior-box is lit.

Creating well-exposed images is challenging. The light, material, and geometry must be tuned to create both realistic and useful exposures. The variety of parameters combined with the physically based renderer leads to a wide variety of exposure between very dark and very bright images. In section 6, we explore a variant of the model which adjusts the exposure dynamically.

**Exterior.** Buildings do not appear in isolation, and their appearance is modulated by the light, shadows, and camera used to image them. Our geometry is lit by emission from the skybox and directly with a sun-lamp. The sun-lamp's light filters through an "urban canyon" that simulates the light passing around other buildings as shown in Fig 8. The geometry allows indirect light paths and shadows to fall onto the building but is not directly visible to the camera or in reflections.

To add variety and realism to the exterior of the building, we add street clutter. This clutter includes objects as varied as trash cans, signs, scooters, and delivery lockers. We collected a set of 278 varied meshes and textures using a hand-held LiDAR and RGB scanner (see Fig 9) which is augmented with a set of images of 5,630 street signs (Anon) with the backgrounds removed. A subset of these were gathered into collections which could be repeated to model such features, including traffic cones, trees, or bollards. The primary requirements for these meshes are that they belong in the street scene, have no personal information (e.g., car number plates) and do not contain building windows (which would degrade labeling accuracy).

Drain pipes and wires are generated by a unified system from a graph of potential edges (PEs). Paths through this graph are extruded to create drainpipe or electrical wire geometry. Using a modification to CGA - we a face can be split to its constituent edges - we build the graph during evaluation of the building grammar. The graph edges include the gutter, and bottom of the walls, as well as vertical and horizontal edges within each rectangle in the wall. We select sources and sinks (e.g., between the gutter and floor level or between two random points in the graph) and find the shortest path between them. These paths are smoothed appropriately (e.g, wires are curved and have much more variability), offset from the wall, have profiles and textures applied, and are added to the scene.

A panoramic skydome adds a background that is often visible in the reflections from the window's glass. This background is selected from a library of street view images. The background may contain other buildings' windows; to avoid degrading the quality of labeling results, this area is unlabeled. A simple circular floor supports the building.

**Camera Position.** The baseline camera positioning switches between two modes: one third of samples position the camera randomly inside a box the width of the house, between 0.5 and 1m from the ground, and 2 to 8m from the facade. The other third are sampled from a position in front of the window - sampled from a $4 \times 2$ meter box parallel to the wall positioned directly in front of the window. This combination was motivated to approximate a held camera, and occasional use of a a higher camera (on a hill or from a second building).

The camera's field of view is computed from the angle between the corners of the window and the camera position. It is perturbed from the largest apparent angle between window diagonals by a normally distributed factor with $\mu = 1.1\sigma = 0.1$. The camera's direction is computed with z-up, pointing towards a target uniformly sampled from a $20 \times 20$cm rectangle in the center of the window.

Occasionally a camera positioned in this way will be behind a large object of street clutter. If 2 out of 5 key points (corners and center) on the window are occluded, the offending clutter object(s) are removed.

**Parameters** The design of the parameter system in a synthetic model has a number of goals. Primarily, it should create distributions of parameters with a good evaluated task accuracy and visual realism.

To achieve these goals within a complex realistic model, it necessary to start with an estimated distribution, which can be iteratively refined with a generate-inspect-update process. While some progress has been made on automated parameter selection (Kar et al., 2019), our large number of parameters (up to 21,735 observed) and non-differentiable renderer makes this challenging; these constraints consequences of our decision to focus on variety (a large number of parameters are required to drive and coordinate a large number of features) and realism (we use a physically based path-tracing render). Therefore, an initial parameter distribution is usually estimated by the engineer or artist who creates the synthetic model. Often these parameters will be estimated on a prototype or incomplete model. A parameter system must therefore support iterative development should track parameter and code changes together their impact on task performance, and be able to explore sequences of changes in the distributions.

The parameter system should be reproducible, in that we must be able to run the same software code-path multiple times when debugging or rendering different variations of the same geometry on a cluster of computers. It should also be robust – small changes in the model's code should have small changes in the output. These requirements together imply that the parameter system has to run in a hybrid mode, with some parameters specified, and others drawn from the distribution. The parameter sampling from any distribution should also be accurate (i.e., as random as possible) over a wide variety of hardware and timings.

The system must support both continuous (e.g., window width, wall texture color) and discrete (e.g., shape of windows, type of window dressing) parameters. The distributions WinSyn uses are typically uniform or Gaussian for continuous parameters and Bernoulli for discrete parameters. But combinations of these quickly become complex when repeated parameter selection moves control to different code branches; these branches may be a function of both parameters and geometry. An example is window-pane splitting - we may continue to split windows until we hit a geometry constraint (they become small) or a parameter decides we should stop (e.g., maximum split depth or stopping early to create more large windows). Another example is that our building modeling language, CGA (Mueller et al., 2006) uses repeated "relative" splits in which a parameter (for example a uniform continuous window panel width parameter) is adjusted to fit an integer number of windows into a wall (a geometric constraint).

We found that apparent realism was enhanced when it was possible to share parameters between disparate parts of the model. For example, to use the same material on a wall as window-frame, balcony base, or drain pipe.

To implement these requirements WinSyn uses several mechanisms in concert to build a parameter system.

- The distributions are defined in the code by their type and parameters (for example mean and deviation for a Gaussian). At runtime, these are are sampled keyed from a unique name.

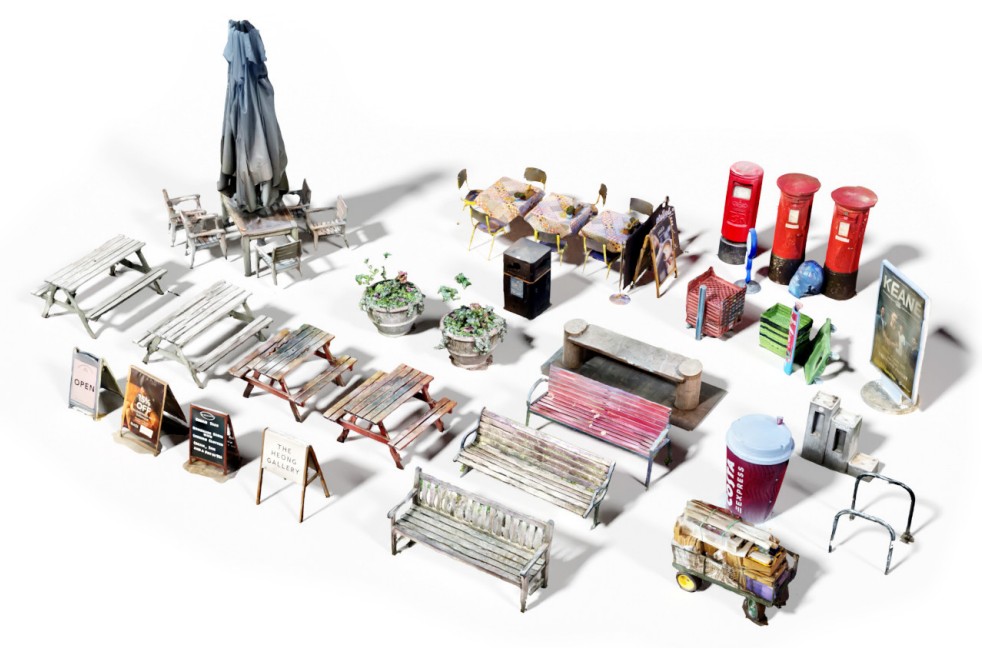

Figure 9: A sample of 27 laser-scanned clutter meshes from our collection of 278, of which 38 are suitable for placement on the wall.

- To ensure the names remain unique between code branches (e.g, for each window independently), we compartmentalize the name spaces into *nodes*. These are stored in a tree parallel to the code branches.

- the sampled parameters are stored in a tree of nodes. This is serialized to disk (as a json file) named for random seed of the root node.

- With some probability distribution, a certain node may differ a sample to its parent. This mechanism support shared parameters across the model.

- If the same model is rerun, we we load the node-tree. As the model is executed we look up values in the tree: found parameters are returned (this provides reproducibility) otherwise they are sampled. We may have to perform mixed lookup and sampling if the code has changed, causing different values to be sampled (this provides robustness).

- As each node is created, a pseudo-random generator is created and initialized by a parameter sampled from the parent node's generator. This compartmentalized random increases robustness, as subsequent samples from the parent generator will not impact the children. The seed for the root node describes the whole scene and is computed as a hash of the current time and compute node.

- We track changes to our distributions through development using version control software and a continuous integration system. This allows us to track accuracy as changes are made.

During the development of WinSyn, we iterated the parameter distributions based on assessed mIoUs 7 on real training data, synthetic data, as well as label integrals 10. Careful examination of these results often reveals features which are under-performing. We found that evaluating labeling accuracy on synthetic hold-outs was useful to validating synthetic labeling and find bugs.

**Rendering.** We use the Cycles renderer (cyc, 2023), with 256 samples per pixel and the default Open Image Denoiser (Intel, 2023) with a $512 \times 512$ pixel resolution to create our color images. The average time to generate a synthetic datum (color image and label map) was 49.8 seconds. The dominant aspect of the rendering was the geometry generation (mean 38.1 seconds), followed by the rendering of the color image (9.2 seconds), and finally the rendering of the labels (2.4 seconds). As in Figure 11 there was considerable variation in these values; we observe the time to generate an image

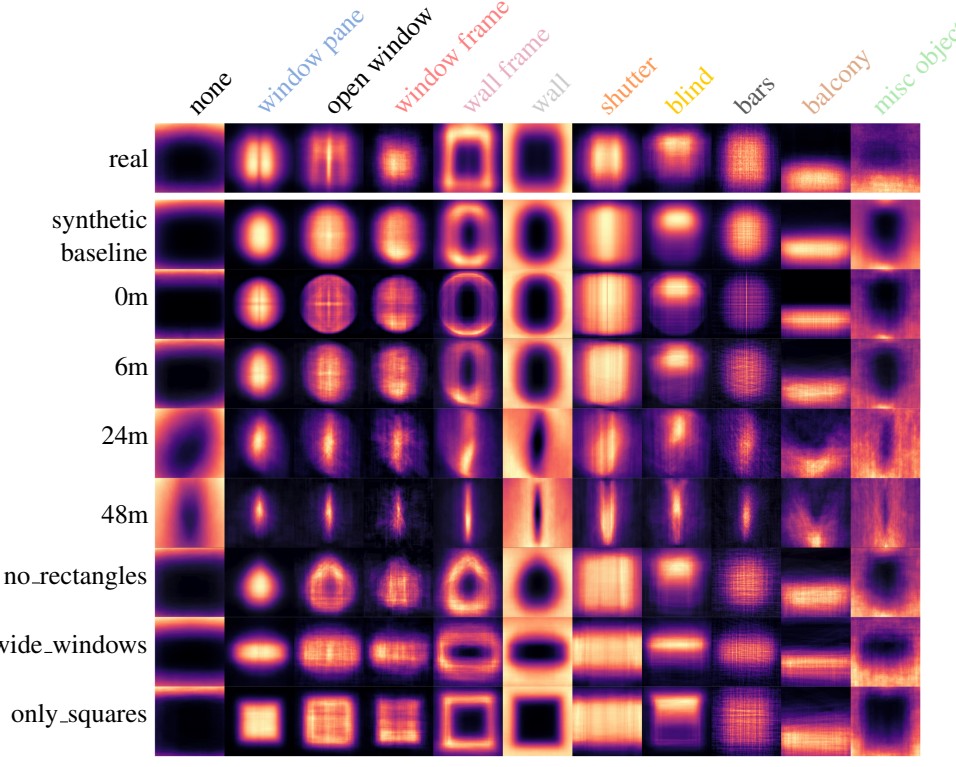

Figure 10: Label integrals for different datasets. For each label (columns) in each dataset (rows) we sum the label masks and normalize per-image. The top row is our ground truth labeled photographs ($n = 9,002$). The second row shows our baseline procedural model ($n = 21,290$). The following rows show various variations ($n = 2,048$) - the *camera location* variations (0m..48m), and *window geometry* experiments (no_rectangles, wide_windows, only_squares).

was dominated by the geometry generation. The hardware used for these timings was a NVIDIA Tesla P100 with a shared Intel(R) Xeon(R) CPUs E5-2699 v3 @ 2.30GHz. This was a single GPU from a multi-GPU machine, so timings may have been affected by other users' workloads. However, it allowed us to distribute work in parallel over 12 nodes, and generate the entire dataset of 21,290 images within a day.

**System.** This synthetic image generation system is implemented in Python 3.10 within the Blender (Community, 2023) 3.3 modeling package. We make use of a number of Blender's features including:

- Screen-space subdivision allows our procedural textures (e.g., brick) to generate geometry as a separate depth channel. This allows the textures to self-shadow (e.g., one brick casting on to a brick below), but can be very GPU memory intensive.

- Geometry-nodes are utilised to features that is are repeated and benefit from shared geometry instances. We use them to create roof tiles, dirt (leaves and litter) on the floor, blinds, and slats. The node-based "language" to describe these is somewhat limiting, but allows faster development and lower memory use.

- Publicly available shaders and geometry nodes-trees. There are a wide number of marketplaces with content available for Blender for free or relatively low cost. However, the licenses may limit the downstream applications and distribution of the model.

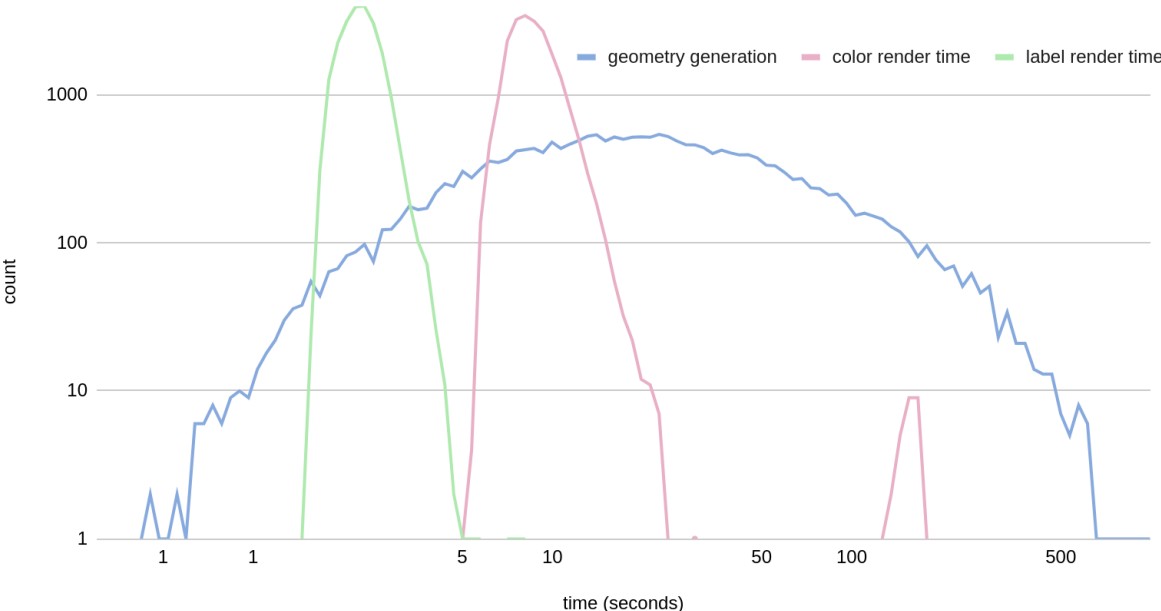

Figure 11: Generation time histogram for the synthetic dataset. Note the log-scale axes as the process was dominated by the geometry generation. Mean total time per sample was 49.8 seconds. The physics simulator ran within the geometry generator for timing purposes.

# 6 VARIATIONS

To demonstrate the applicability of WinSyn on real to synthetic we investigate the impact of a variety of variations on labeling performance. We create 59 datasets, each of size $n = 2,048$, as variations of our synthetic data with different sizes and mixes of real or synthetic data, geometry, textures, lighting, labels, and camera positions; these are tested on our standard labeled data split $n = 4.9k$. We provide analysis of these results to provide guidance to others creating synthetic procedural models. In this section we describe how each experiment varied from the baseline.

## 6.1 MATERIALS

Four of the the material variations use no lighting model (labels, albedo, normals, and lines). The remainder use a simple diffuse lighting model, with no direct sun. See Figure 12. The geometry is identical for each render. The material variations studied are:

- *labels*: The labels rendered from the synthetic dataset. These were included as a baseline for a "bad" variation with label cohesion, but very low realism and little in common to the real data.
- *albedo*: The albedo pass from the renderer.
- *normals*: The screen-space normal map.
- *vornoi_chaos*: Each object has the same 3D vornoi-cell texture, but with different scale and offset parameters. Each cell has a random color.
- *monomat*: For each object type in the scene, we apply the same material across the whole dataset. The parameters for the procedural materials are fixed.
- *col_per_object*: Each object in the scene is a random color.
- *diffuse*: The entire scene has a single mat material with gray color.
- *edges*: An edge renderer is used to create a sketch-like image of the scene.
- *texture_rot*: For each object we select one from collection of 50 geometric textures, and apply a random lighting and scale.

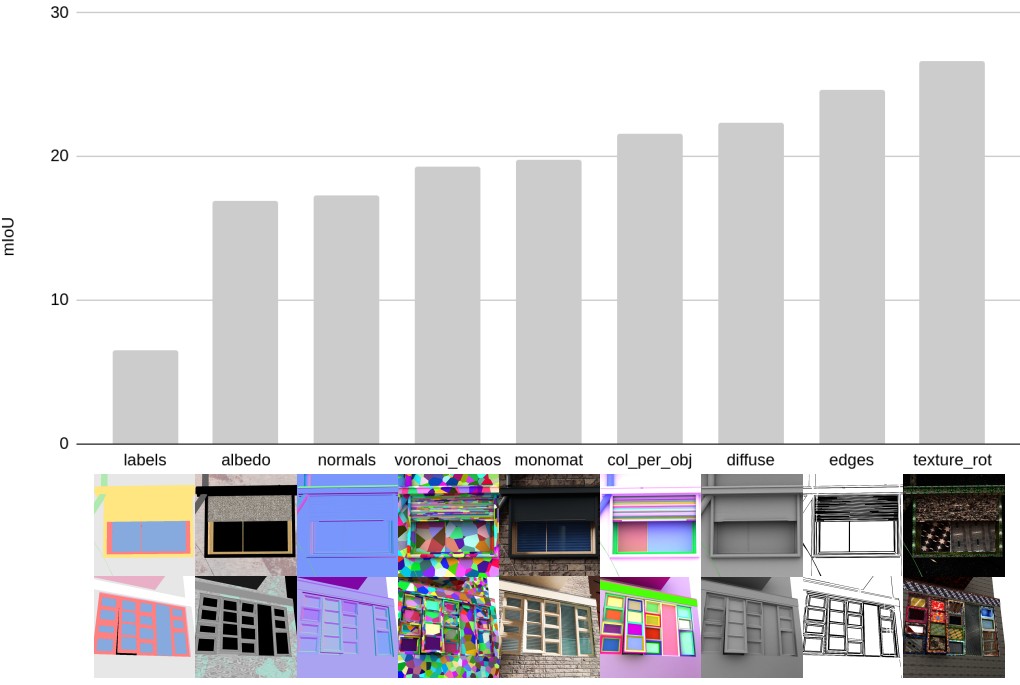

Figure 12: The impact of a variety of different materials on labeling task performance.

## 6.2  Lighting Models

The results from the lighting variations are shown in Figure 13. Each variation explores a different aspect of the lighting model. The variations modeled are:

- *albedo*: (as the Material variation) The albedo pass from the renderer. No other lighting model.
- *phong_diffuse*: The diffuse term from the Phong lighting model with a gray material. No other lighting model.
- *diffuse*: (as the Material variation) The entire scene has a single matt material with gray color.
- *night_only*: Only the night (less light from the sun, brighter internal lighting) mode is used. Dark images.
- *no_sun*: No directional light source in the scene.
- *no_bounce*: The path tracer terminates the trace after the first bounce.
- *fixed_sun*: The sun is always in the same location and size.
- *day_only*: Only use the day lighting model (no night).

The usual whitening across a dataset is performed on all variations before training and testing; but this post-process reduces useful color depth and is dataset-wide. For these lighting experiments, we use also a per-image *exposure* pass within the render pipeline to preserve color depth and perform brightness equalization. This is similar to histogram equalization in that it adjusts the image brightness based on the central areas of the image, simulating the auto-exposure mechanism in a digital camera. We can see that for poorly exposed dataset (such as *nigh_only*) this significantly improves the performance. However for well balanced datasets (*day_only*) there is a slight performance degradation (Figure 13).

Generally, it is important to have realistically exposed images. The synthetic model has a night-time mode under the assumption that windows would look very different when lit from within; however the night-time lighting setup was counter-productive, and the task effectiveness is improved without it. However, having a sun with multiple positions creating shadows was somewhat useful to the task.

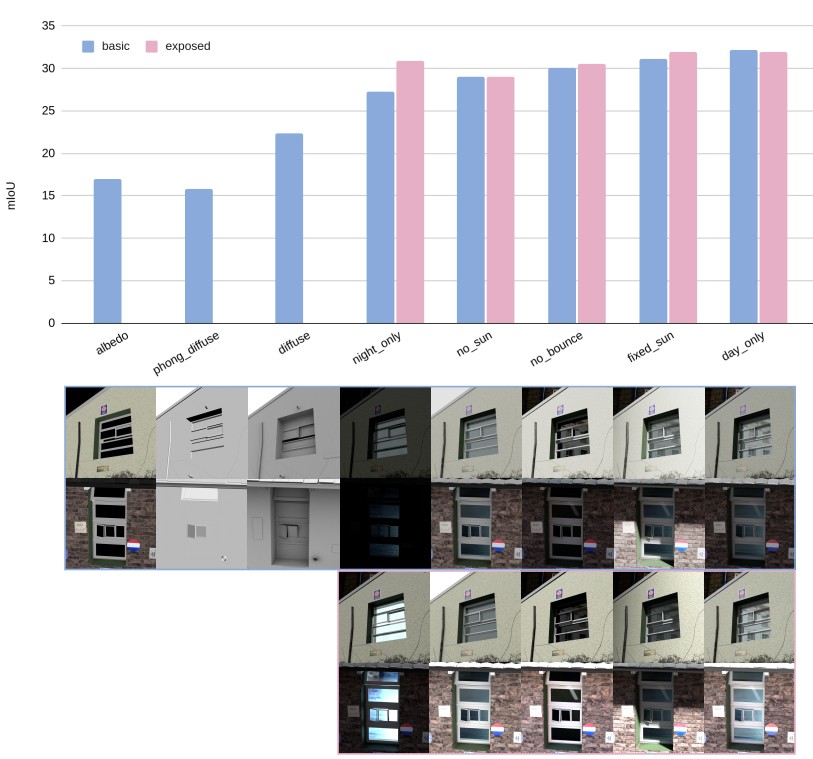

Figure 13: The impact of different lighting models on the labeling task mIoU.

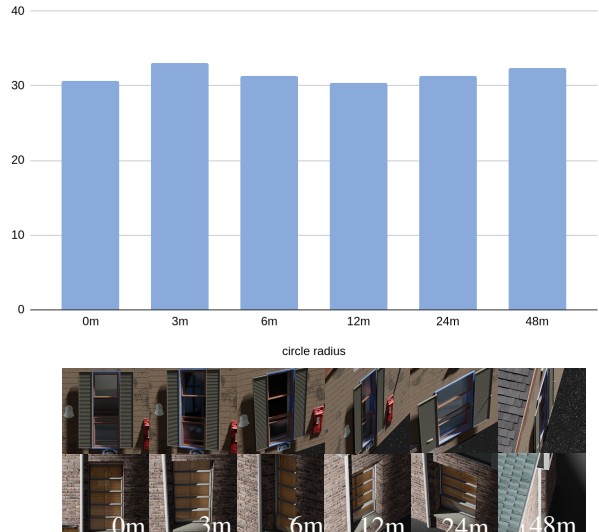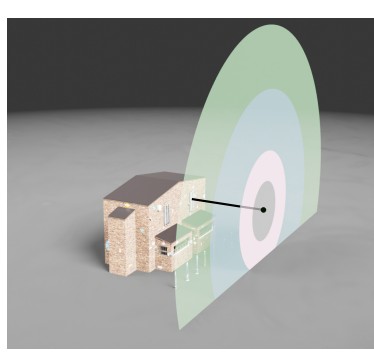

Figure 14: Left: the mIoU and average accuracies for different camera-sampling-circles. Right, top: The position of the circles' centers is $5m$ (black line) front of the top right window. For scale, the green circle has $r = 12m$. Bottom left: samples from the different distances.

## 6.3 CAMERA POSITION

Windows are a useful domain for studying camera position distribution as they have an obvious unambiguous canonical orientation. In this variation we experimented with the distribution of camera positions. We used a simple model which sampled a camera position over a circle, of radius $r$ meters, truncated at the floor plane. The circle is positioned 5 meters from the wall, directly in front of the window. See Figure 14. As $r$ increases, the majority of the circle area moves away from the window, and so the camera angle to the wall becomes very shallow in many samples. Larger circles have lots of area higher up, so create unusual camera angles not present in the photographic dataset. However we observe that high $r$ does not significantly strongly impact the segmentation task accuracy; even at extreme camera angles. This contradicts our assumption when building the model that camera location distributions should be guided by the label integrals, which clearly show a large divergence for wide camera angles as illustrated in Figure 10: $48m$.

The baseline models removes clutter from the scene which occludes the camera view of the primary windows. These variations do not perform this occlusion check. At more extreme camera positions, there is a greater chance of objects (clutter, bay windows, or balconies) block the view.

## 6.4 LABELS MODELED.

Real-world diversity of even simple man-made objects such as windows is huge (as Figure 1). When developing a procedural model as a synthetic data generator, we must prioritize where to spend software development effort. Our model was developed approximately in order of label sizes: largest first (Figure 5). In this variation, we take the completed model, and perform ablations by removing each class in turn to study the impact of each class in a systematic manner. From the results Figure 15 we observe the diminishing returns in mIoU for the later labels, as the features get smaller and the larger choice of labels makes labeling harder.

The label levels variations are:

- *lvl1*: Only the wall geometry is present.
- *lvl2*: Add a window panes. It is not recessed.
- *lvl3*: Add a wall-frame and recess the window pane.

- *lvl4*: Add a window-frame and any splits to the window panes.
- *lvl5*: Add window shutters.
- *lvl6*: The balconies are added. These use the *balcony* for their guard rails and *wall-frame* labels on their lower parts.
- *lvl7*: Wall and floor clutter is added to the scene.
- *lvl8*: Add window blinds.
- *lvl9*: We finally add the smallest label, bars, to the dataset.

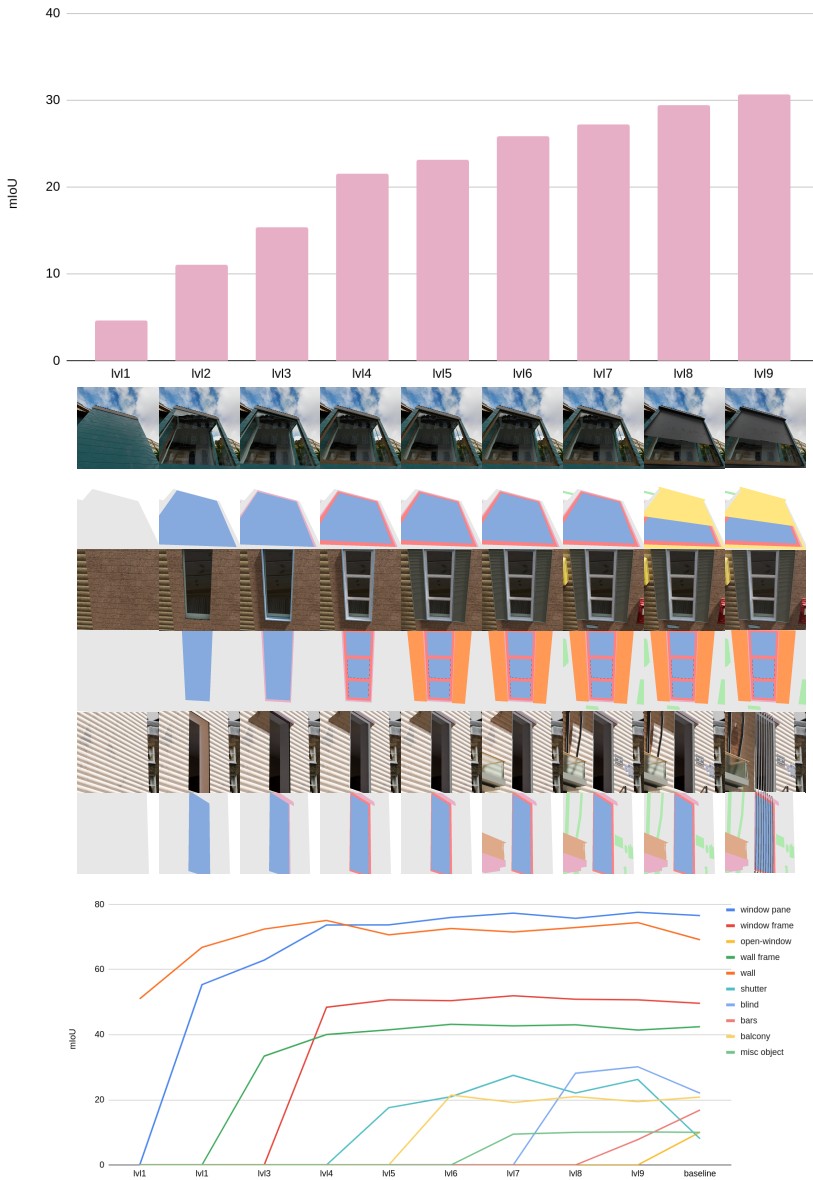

Figure 15: Top: The cumulative impact of modeling a different labels. Beginning with only walls (label level 1 - *lvl1*), adding window panes (2), wall-frames (3), window-frames (4), shutters(5), balconies (6), misc (7), blinds (8), and bars (9). The difference between lvl9 and the complete baseline model is the addition of interior dressing (curtains), open windows, and windows without glass. Middle: examples from the different levels. Bottom: the impact on the per-class IoU of each label. We note that sometimes adding more labels will harm segmentation quality.

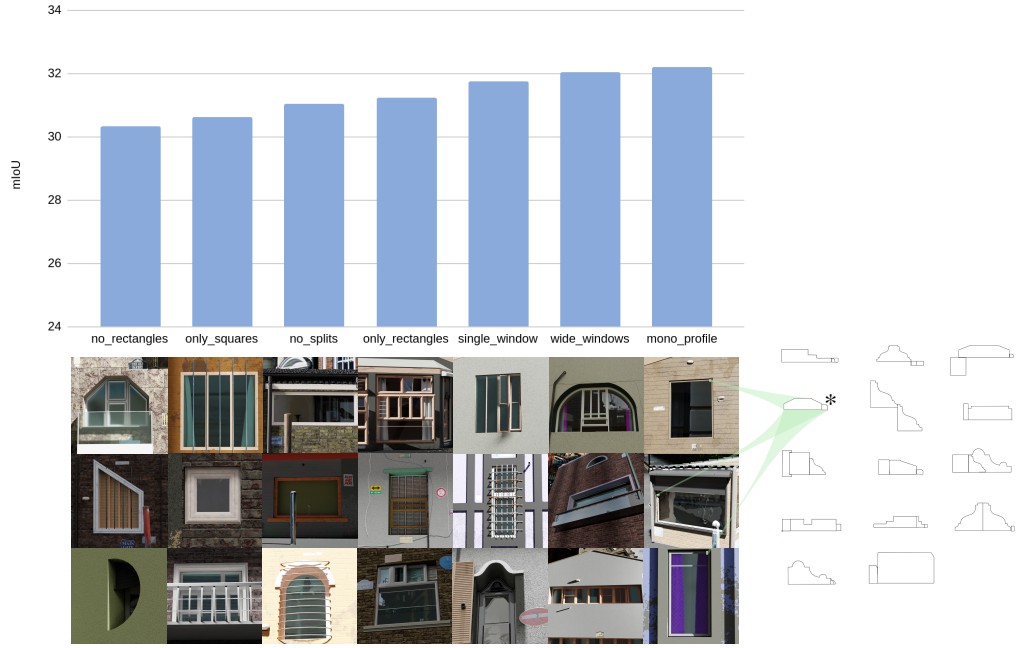

Figure 16: Top: The impact of changing the window shape and geometry on the mIoUs of the labeling task. Bottom left: examples of the $n = 2,048$ training examples in each variation. Bottom right: the full set of window-frame profiles for the baseline model; an asterisk marks the single profile used in the *mono_profile* variation. Note vertical axis starts at non-zero.

**Window Geometry**

Here we study the impact of changing the distribution of parameters which impact the window geometry, as Fig 16. Note we do not have a uniform base geometry in this experiment as the distribution changes required a set of window parameters together. The label integrals for some of parameter distributions are shown in Figure 10. The window geometry distributions variations are:

- *no_rectangles*: Only non-rectangular windows (circular, arched, and angled)
- *only_squares*: Only square windows.
- *no_splits*: No window-pane subdivision took place.
- *only_rectangles*: Only rectangular windows.
- *single_window*: Only a single window created.
- *wide_windows*: The width/height parameters are adjusted to create more wider, shorter windows.
- *mono_profile*: We reduce the number of all the extruded profiles in the system. These are used on window frame, sill, lintels, roof gutters. The set of profiles used for windows is reduced from the baseline (Fig 16, right) to a single profile.

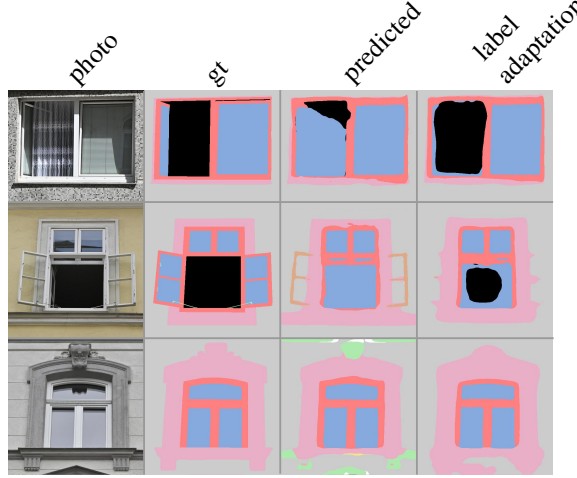

Figure 17: Example outputs from our label adaptation (Wood et al.) label-to-label network. We note that the network is able to identify likely mislabeling (such as an open window without the *open-window* label) and correct the output appropriately.

## 7  LABEL ADAPTION

As Figure 17, we observe that improving labeling by just learning from the labels provides a powerful supervised prior to improve the accuracy between dataset. This semi-supervised technique is able to identify common problems with the synthetic-trained labeling network and correct them.