# OpenReview forum: "WinSyn: A High Resolution Testbed for Synthetic Data"
_ICLR.cc/2024/Conference — ICLR 2024 Conference Withdrawn Submission_

### Official Review · Reviewer_ZJ39 · 2023-10-18

**Soundness:** 3 good
**Presentation:** 3 good
**Contribution:** 1 poor
**Rating:** 3
**Confidence:** 3

**Summary:**

The paper presents "WinSyn," a dataset designed for sim-to-real research, consisting of high-resolution images of building windows from various global locations and an additional set of synthetic images generated from a photorealistic model. The dataset aims to advance the study of sim-to-real transfer. The context of window designs is chosen for their diverse geometry and materials. Some experiments are conducted using a segmentation model to analyse the impact of various factors on the generalization performance.

**Strengths:**

### Strengths

1. Motivation:
I do see the reasoning behind creating the dataset i.e. to futher study the sim-to-real gap in a controlled setting. The scale and type of data (including RAW images) suggest that significant effort has been invested in creating the dataset. While the paper primarily focuses on synthetic data for machine learning, the inclusion of other labels in the dataset can make it useful for other tasks (like the ones mentioned in the paper-superresolution, 3D reconstruction, and generative modeling). Exactly how it can be used for these tasks is not explained.
2. Quality: I like how the paper systematically explores the effect of variations on the segmentation accuracy (/mIoU). This rigorous analysis of the effects of data variation can lead to some interesting insights but I'm not sure of how much practical use these are.
3. Clarity: The introduction gives a good overview of the main contributions of the paper: the proposal of a new dataset and an exploration of the synthetic to real performance gap, and the importance of research in synthetic data generation. The description of the problem and the dataset is quite clear and easy to follow. I also appreciate the fact that the paper acknowledges is clear about its limitations, particularly that the gap between synthetic and real data still remains unexplained.
4. Significance: The dataset, as described, is positioned as a benchmark that could aid other researchers in studying the sim-to-real problem, and I acknowledge that benchmarks have been vital in advancing ML research.

**Weaknesses:**

### Weaknesses

1. Experiments:
A main motivation for the dataset is that it can help to address the gap between synthetic and real data, however the paper doesn't really show how the dataset can be used to explain why this gap exists. I would suggest that the authors dive deeper into potential reasons for this gap. Are there features or nuances of real-world images that the synthetic data does not capture (perhaps gradient attribution or some other attribution method can be used to explore this).
2. Originality and contribution:
While the paper briefly mentions competing datasets, it doesn't give a clear comparison between these. Understanding how this work differentiates or improves upon existing dataset is crucial for assessing its novelty and significance. There are many synthetic and real dataset on useful real-world domains such as street scenes for autonomous driving (eg. CARLA/SynthIA/GTA5 vs KITTI/cityscapes). A dedicated section comparing the new dataset and procedural model to existing ones in terms of size, quality, versatility, and applications would enhance the paper's depth and relevance.
3. Lack of demonstrating broader applications:
The paper loosely mentions applications like superresolution, 3D reconstruction, and generative modeling but does not provide experiments or detailed insights into how the dataset enhances these applications. Detailed experiments or use-cases showing the effectiveness of the dataset in these applications would give readers a tangible understanding of its real-world significance.
4. Addressing the sim-to-real gap:
The authors emphasize the need to understand critical differences between real and synthetic data, but no conclusive insights are drawn about this using the dataset. A dedicated analysis or experiment that can help to highlight and understand the differences between the domains would add substantial depth to the paper.

**Questions:**

### Specific questions:

1. From the experiments in the paper it seems it's quite difficult to draw conclusions as to what is leading to the synthetic to real gap. Do you believe the dataset can actually be used to draw such insights and how do you think this could be done?
2. The dataset seems to be a completely "toy" domain that might also be quite expensive to train. I would therefore be a bit concerned about researchers wasting hours of compute (with corresponding environmental impact) on models that aren't useful for any other tasks. How would the authors address this?

---

### Official Review · Reviewer_YaDy · 2023-10-31

**Soundness:** 3 good
**Presentation:** 3 good
**Contribution:** 3 good
**Rating:** 5
**Confidence:** 4

**Summary:**

The paper introduces a dataset of synthetic (21,290 procedurally generated, high definition) and real images (75,739) of windows for semantic segmentation and related tasks. Real images are collected globally (28 locations) and cropped to window regions, resulting in 89,318 crops, of which about 10% (9002) are labelled with dense segmentations.

The paper also proposes a procedural model for window generation. Compared to existing datasets, the proposed dataset focuses on a single class (windows), but considers many associated labels (e.g., wall, window page, blind etc. to a total of 11 labels including ‘unlabelled’).
Various experiments are conducted comparing semantic segmentation performance of real and synthetic data:

*	When little real data is available, synthetic data improves segmentation (IoU), but when lots of real data is available, synthetic data reduces performance
*	Samples per pixel (spp) in rendering synthetic data is important for how effective synthetic data is.
*	Label adaptation (dynamically improving labels with a network trained on real and synthetic pairs) improves performance
*	Significant performance variations exist depending on where (geographically) did the training and testing data come from.
*	An ablation study across rendering parameters (lightning, camera, geometry and others) is provided.

**Strengths:**

The paper collects a large and diverse dataset with per-pixel annotations that are consistent across synthetic and real examples. Such as dataset facilitates research into development of techniques associated with the use of synthetic data.

**Weaknesses:**

Parts of the paper are difficult to follow. The introduction could be streamlined. There is no summary of key contributions and findings/results obtained using the proposed synthetic data.

It is mentioned that in addition to semantic segmentation, dataset may include depth (see Figure 1), but no information about depth or any results using these annotations are provided.

Finally, a summary table comparing properties of the proposed dataset and related datasets is missing.

**Questions:**

From the information provided, it is not clear what gap in ML datasets/research does the proposed dataset address. Do we expect findings from analysis of windows to generalize to more general semantic segmentation segmentation tasks?

Please also provide specific details of what labels does the dataset include and what ML tasks it can be used for.

---

### Official Review · Reviewer_7VZc · 2023-10-31

**Soundness:** 1 poor
**Presentation:** 2 fair
**Contribution:** 2 fair
**Rating:** 3
**Confidence:** 3

**Summary:**

The paper proposes a data with a combination of synthetic data and real data of windows, to facilitate research in methods including segmentation and domain adaptation. The paper includes exhaustive description on the specifics of the proposed dataset, as well as benchmark on segmentation under various training/testing strategies/splits to demonstrate the domain gap between real/synthetic splits and splits of different geo locations. The paper also evaluates the impact of synthetic dataset variations on segmentation, including rendering/design/labeling choices.

**Strengths:**

[1] The paper proposes a dataset with both synthetic/real splits on a specific domain, i.e. windows, with high quality renderings and real captures, as well as detailed statistics, benchmarks on the task of segmentation.

[2] The paper elaborately justifies the choice of windows as the domain to model in the proposed dataset, based on its intra-class variation across geographical locations and appearances, availability of source images, and suitability for tasks including segmentation and domain adaptation.

**Weaknesses:**

[1] Limited contribution. The proposal of a new dataset including both synthetic and real data is always appreciated; however, the paper does not fully justify its significance over other existing datasets. For instance, with respect to the application of segmentation, there are a wide range of datasets especially on city scenes or driving scenes that have been extensively utilized for the task and even in the context of domain adaptation. In this paper the choice of windows as the domain is properly justified, but it did not answer the question on whether existing datasets can already serve the same purpose (or more). In this case, the significance of the dataset is in question.

[2] Limited evaluation. The paper evaluates the proposed dataset in segmentation tasks on various settings. However a missed opportunity given the availability of both real and simulated splits and the discovery of domain gap in its evaluation, is to test the dataset on domain adaptation methods. The paper could have tested the dataset with existing domain adaptation methods on what the baseline benchmark looks like, potential advantage or challenges of the dataset when compared to other existing datasets in the context of domain adaptation. Otherwise it is difficult to evaluate the impact of the proposed dataset on domain adaptation research, without including actual evaluation results.

[3] Writing. The paper includes lengthy introduction and justification of the choice of windows, in the Introduction section. The text mostly served its purpose. However the writing can be greatly improved. For instance, reorganizing the first paragraph of the Introduction section from 6 points into an organic flow of arguments can be made: the main task the dataset is motivated towards, issues with existing datasets, features of the proposed dataset which significantly differentiates itself from existing ones, and other major features of the proposed dataset. Additionally, the 6 points can overlap: 1st and 4th points both talk about why choosing the domain of windows, and also overlaps with the entirety of the second paragraph. The 6th point is confusing: any dataset comprised of images can be used to extract image features. Finally, additional introduction of Wood et al. before referring to it may benefit the readers in better understanding the last paragraph of the Related Works section and Label Adaptation of Section 6.

**Questions:**

[1] Can existing domain adaptation methods (e.g. GeoNet and related methods: https://tarun005.github.io/GeoNet/) be applied to the proposed dataset, and what are the observations compared to being applied to other existing datasets?

[3] Can other applications be enabled by the proposed dataset besides segmentation?

**Details Of Ethics Concerns:**

A team of photographers were hired to take photos of windows in the streets; as a result there is a good chance that some of the original photos may be related to privacy concerns especially those taken of private residences. The paper mentions that such samples are properly filtered in the final dataset; however more details are to be inspected: are the privacy-concerning raw images properly destroyed? Do any such photos end up in the final dataset and how to guarantee that there are none?

---

### Official Review · Reviewer_CUsR · 2023-11-02

**Soundness:** 2 fair
**Presentation:** 2 fair
**Contribution:** 1 poor
**Rating:** 3
**Confidence:** 4

**Summary:**

This paper proposes a unique window dataset, which contains 75,739 high resolution real images and 21,290 synthetic images created by procedural modeling. 9002 of the real images contain detailed segmentation masks. Authors then trained semantic segmentation network on different subsets of this dataset with different settings, to study which factor of the dataset may impact the segmentation accuracy.

**Strengths:**

1. This paper proposes a unique window dataset, which contains 100K high quality window images with close to 10K high-quality semantic segmentations. This dataset may be useful for specific window related applications.

2. Authors make sure that all real and synthetic images won't have license issues, which will be beneficial for the computer vision and machine learning community to adopt this dataset.

**Weaknesses:**

1. It is difficult to say if window category is representative. Author may need statistic or experimental proof to show why we should choose window to study synthetic to real or generative modeling. It will be good to show that the conclusion we find in this paper can also be applied to more complicated scenes.

2. Some claims in the paper are not verified through experiments. While authors mention generative models and synthetic-to-real in the abstract and introduction, all experiments are on semantic segmentation and we do not have concrete evidence if this dataset can be used for generative modeling or synthetic-to-real domain transfer.

3. The size of the dataset may still be limited. The whole dataset contains less than 100K images of windows, which is still much less than typical dataset used for semantic segmentation (COCO) and generative learning (LION). It is difficult to say if the conclusions from experiments on this dataset can be generalized.

**Questions:**

My most major questions is that given the size and the limited variation and size of this dataset, it is difficult to say if the conclusions on domain transfer, semantic segmentation or generative learning we find on this dataset can be generalized to other more broad setting. It may be better to focus on window-related application, such as reflection removal or even geometry reconstruction of transparent/reflective surface

**Details Of Ethics Concerns:**

As far as I know, there is no ethic concern.